# Low-voltage 2D materials-based printed field-effect transistors for integrated digital and analog electronics on paper

Silvia Conti [1,7], Lorenzo Pimpolari[1,7], Gabriele Calabrese [1], Robyn Worsley[2], Subimal Majee[2], Dmitry K. Polyushkin[3], Matthias Paur[3], Simona Pace [4,5], Dong Hoon Keum[4,5], Filippo Fabbri[4,6], Giuseppe Iannaccone[1], Massimo Macucci[1], Camilla Coletti [4,5], Thomas Mueller[3], Cinzia Casiraghi[2] & Gianluca Fiori[1✉]

Paper is the ideal substrate for the development of flexible and environmentally sustainable ubiquitous electronic systems, which, combined with two-dimensional materials, could be exploited in many Internet-of-Things applications, ranging from wearable electronics to smart packaging. Here we report high-performance $MoS_2$ field-effect transistors on paper fabricated with a "channel array" approach, combining the advantages of two large-area techniques: chemical vapor deposition and inkjet-printing. The first allows the pre-deposition of a pattern of $MoS_2$; the second, the printing of dielectric layers, contacts, and connections to complete transistors and circuits fabrication. Average $I_{ON}/I_{OFF}$ of $8 \times 10^3$ (up to $5 \times 10^4$) and mobility of $5.5$ cm$^2$ V$^{-1}$ s$^{-1}$ (up to 26 cm$^2$ V$^{-1}$ s$^{-1}$) are obtained. Fully functional integrated circuits of digital and analog building blocks, such as logic gates and current mirrors, are demonstrated, highlighting the potential of this approach for ubiquitous electronics on paper.

[1] Dipartimento di Ingegneria dell'Informazione, University of Pisa, Pisa 56122, Italy. [2] Department of Chemistry, University of Manchester, Manchester M13 9PL, UK. [3] Institute of Photonics, Vienna University of Technology, Vienna 1040, Austria. [4] Center for Nanotechnology Innovation @NEST, Istituto Italiano di Tecnologia, Pisa 56127, Italy. [5] Graphene Labs, Istituto Italiano di Tecnologia, Genova 16163, Italy. [6] CNR, Scuola Normale Superiore, Pisa 56127, Italy. [7] These authors contributed equally: Silvia Conti, Lorenzo Pimpolari. ✉email: gianluca.fiori@unipi.it

In recent years, electronics has witnessed impressive technological achievements, owing to the development of new processes and materials with extraordinary electrical and mechanical properties, which have enabled the development of Internet of Things applications, ranging from wearable electronics to mobile healthcare. This has led to a continuous and marked increase in demand of light-weight, flexible, and low-cost devices, posing strong constrains on traditional fabrication methods[1,2]. In addition, this type of pervasive and versatile electronics had led to further concerns on sustainability, such as the treatment of waste at the end of the product life-cycle. Derived from abundant and renewable raw materials, paper-based consumer electronics is expected to alleviate landfill and environmental problems and to reduce the impact associated with recycling operations, whilst offering cost-effectiveness and large flexibility[3]. Despite the fact that several devices and applications have been reported in the literature[4], paper is still a challenging substrate for electronics, rarely employed without the addition of coating/laminating layers[5,6]. Its porous structure (which in turn leads to high roughness), limited stability and durability (mainly due poor thermal and humidity resistance), and high hygroscopicity (which can influence the electrical characterization of devices fabricated on top of it), combined with the lack of winning reliable fabrication techniques, is preventing its exploitation at the industrial level[7,8].

Two-dimensional materials (2DMs) combine good tunable electronic properties with high mechanical flexibility, making them extremely promising as building blocks for flexible electronics[9,10]. Moreover, they can be easily produced in solution with mass scalable and low-cost techniques, such as liquid-phase exfoliation[11], enabling their deposition by simple fabrication techniques such as inkjet printing[12–17]. 2D semiconducting materials, such as transition metal dichalcogenides (TMDCs)[18,19], with extended bandgap tunability through composition, thickness, and possibly even strain control, represent promising materials as channels for field-effect transistors (FETs), which are fundamental components in electronics. However, up to now, fully printed TMDC-based transistors have demonstrated limited performance, showing mobility of the order of under 0.5 cm$^2$ V$^{-1}$ s$^{-1}$ and $I_{ON}/I_{OFF}$ ratios of hundreds, using liquid electrolytes as insulating layers[20,21]. Among the various TMDCs, molybdenum disulfide (MoS$_2$) has been widely studied, owing to its outstanding electrical and optical properties[22–26]. Lin et al. [27] reported FETs made with solution-processed MoS$_2$, showing remarkable performance (average mobility of ~7–11 cm$^2$ V$^{-1}$ s$^{-1}$), but device fabrication required acid cleaning and annealing above 200 °C, which are incompatible with substrates such as paper. A large mobility of 19 cm$^2$ V$^{-1}$ s$^{-1}$ for a MoS$_2$/graphene transistor was reported in ref. [28]: graphene allows increasing carrier mobility, but this negatively affects the $I_{ON}/I_{OFF}$ ratio.

We combine chemical vapor deposition (CVD), for the growth of high-quality MoS$_2$ channels, with inkjet printing[29,30], which allows to design and fabricate customizable devices and circuits exploiting 2DMs-based inks, whose capability to be printed on top of CVD-grown materials has been successfully demonstrated in ref. [16]. In this work, an application-specific integrated circuit design approach, known as "channel array", is proposed: this is based on the transfer of strips of CVD-grown MoS$_2$, onto paper substrate where the rest of the devices and circuits, source and drain contacts (which define the effective channel length and width), gate dielectric, gate contacts, and connections, are fully customized exploiting inkjet printing technique, giving a degree of freedom to the designer. This method allows to keep the flexibility and versatility of an all-inkjet technology, with the difference that here a high-quality channel is already placed on the substrate, by taking advantage of the CVD-grown TMDC. Moreover, as both methods are compatible with large-area

fabrication processes, their combination could open a possible exploitation at the industrial level.

The MoS$_2$ FETs fabricated with the channel array method operate at supply voltage below 2 V, with remarkable transistor performance, such as an average field-effect mobility of 5.5 cm$^2$ V$^{-1}$ s$^{-1}$ (with best performance reaching 26 cm$^2$ V$^{-1}$ s$^{-1}$), negligible leakage currents (smaller than 5 nA), and an average $I_{ON}/I_{OFF}$ ratio of $8 \times 10^3$ (up to $5 \times 10^4$). We further exploit the possibility to produce high-performance transistors with the channel array method by demonstrating more complex circuits, such as logic gates (such as NOT and NAND) and analog circuits. This paves the way towards the introduction of the channel array approach in all applications where flexible and/or disposable electronics is required.

## Results

**Fabrication of MoS$_2$ FETs on paper.** The rationale of our approach is the combination of two fabrication techniques, which up to now have been used for very different applications, to have high-quality semiconducting substrates easily customizable to obtain devices and circuits with a versatile printing technique. The advantage of inkjet-printing is the fast prototyping, which allows for on-the-fly corrections as well as easy pattern changes, simplifying the manufacturing process. Moreover, being an additive and mask-less method, it also cuts down materials and energy consumption, reducing the number of processing steps, time, space, and waste production during the fabrication. On the other hand, inkjet-printing presents critical aspects, such as the need to use inks with specific rheological properties, and, more importantly, the current lack of semiconducting 2DM-based inks for high-performance FETs. Even if expensive, lacking in compatibility with arbitrary substrates, suffering from atomic vacancies and batch-to-batch variations, CVD is, so far, the most-promising bottom–up approach to obtain high-quality semiconducting layer and may become the method of choice, also considering the recent progress in the CVD growth of MoS$_2$ involving a low-cost, large-area roll-to-roll approach[31].

Figure 1a illustrates the procedure followed to pattern CVD MoS$_2$ and its transfer to paper substrate (a detailed description of the process is reported in Methods, an alternative method for CVD growth and transfer is presented in Supplementary Note 1). After the transfer, the polystyrene carrier film is dissolved in toluene, resulting in MoS$_2$ strips on paper, as shown in Fig. 1b (an atomic force microscopy micrograph of the MoS$_2$ film on the sapphire substrate before the transfer is reported in Supplementary Note 2).

To evaluate the crystalline quality of the MoS$_2$ before and after the transfer process from the rigid substrate to the paper, Raman spectroscopy is employed. Figure 1c shows the Raman spectra before (red line) and after (cyan line) the transfer. The red spectrum presents the E$_{2g}$ and A$_{1g}$ modes at 383 cm$^{-1}$ and at 403 cm$^{-1}$ of single-layer MoS$_2$, representative of the in-plane and out-of-plane vibrations of S–Mo–S, respectively[32]. After the transfer process, the MoS$_2$ Raman modes appear slightly shifted and broadened, i.e., the E$_{2g}$ and A$_{1g}$ modes peak at 380 cm$^{-1}$ and at 400 cm$^{-1}$, respectively. As previously reported in ref. [33], the softening of Raman modes can be attributed to uniaxial strain, albeit the E$_{2g}$ mode should suffer a larger shift compared with the A$_{1g}$ mode. In our case, the softening of the Raman modes is comparable, ruling out any strain effect on the MoS$_2$. Therefore, we argue that the softening is mainly owing to heating effects related to the poor heat dissipation of the paper substrate. This hypothesis is also supported by the broadening of the full-width-at-half-maximum (FWHM) of both modes. Indeed, the E$_{2g}$ FWHM increases from ~3 cm$^{-1}$, before transfer, up to ~7 cm$^{-1}$ after the transfer process. In the case of the A$_{1g}$ mode, the

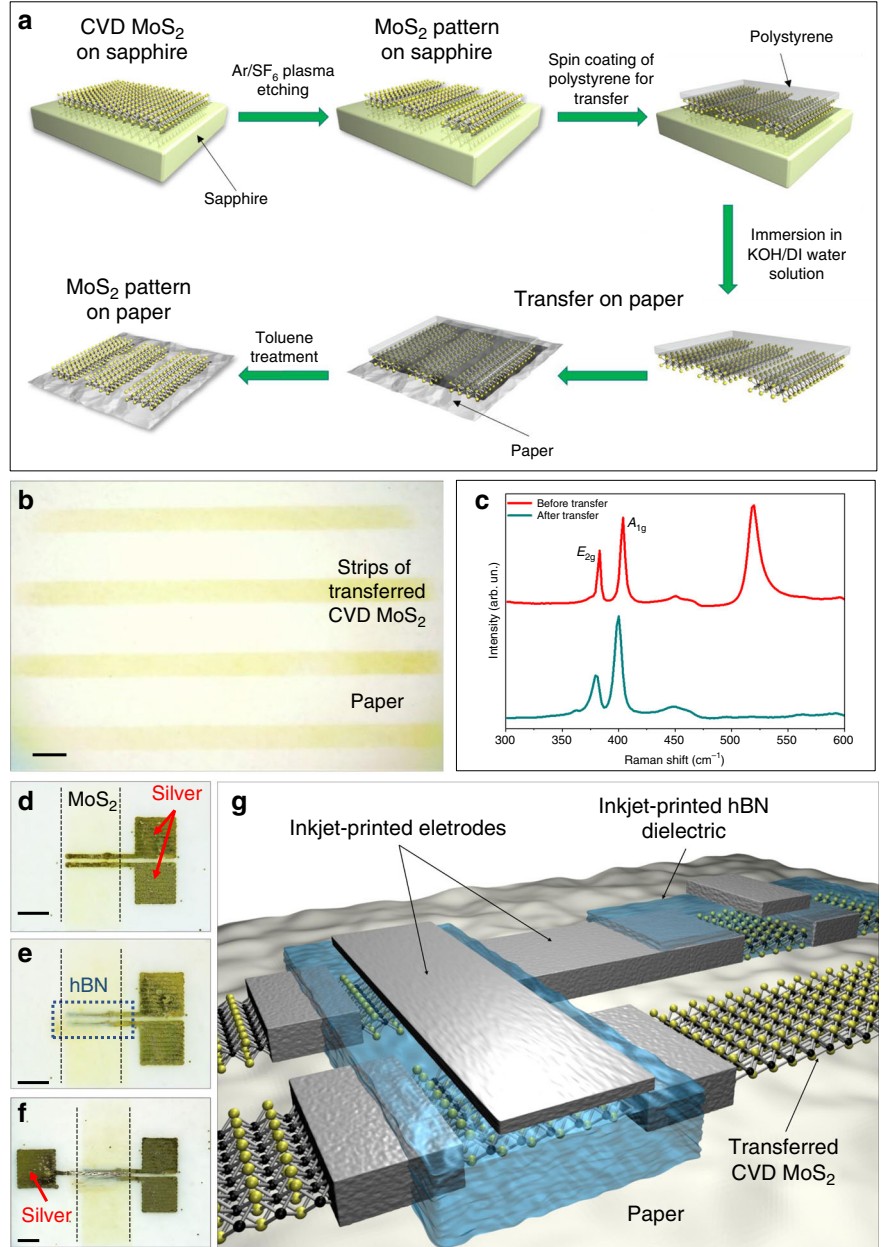

**Fig. 1 Transferring scheme of MoS₂ channel stripes and fabrication process of MoS₂ FETs. a** Schematic representation of the patterning and transferring procedure employed to obtain MoS₂ strips on paper. **b** Optical micrograph showing the transferred MoS₂ strips on paper. The scale bar corresponds to 1 mm. **c** Raman spectra acquired on the as-grown MoS₂ layer on rigid substrate (red line) and after MoS₂ transfer to paper (cyan line). **d**–**f** Fabrication steps of the inkjet-printed transistors on paper: **d** Inkjet-printing of silver source and drain contacts. **e** Inkjet-printing of the hBN dielectric layer (defined by the blue-dotted frame). **f** Inkjet printing of silver top-gate contact. The scale bars in **d**–**f** correspond to 250 μm. **g** Sketch showing an inkjet-printed circuit on paper with CVD-grown MoS₂ channel.

broadening is less evident, with the FWHM increasing from ~4 cm⁻¹ up to ~6 cm⁻¹.

Figure 1d–f show the fabrication of inkjet-printed MoS₂ FETs on paper. First, the source and drain contacts are printed on a MoS₂ stripe to define the channel area of the transistor (Fig. 1d). Second, a hexagonal boron nitride (hBN) film is printed on the MoS₂ channel (Fig. 1e). This 2D insulating material is chosen because of its notable dielectric properties and negligible leakage current[15,20,34,35]. Finally, a top-gate electrode is printed on top of hBN (Fig. 1f). Either silver or graphene inks have been used to print the source and drain contacts as well as the top-gate contacts. The FETs are then connected to each other using the

routes defined between the MoS₂ strips, to create the integrated circuit in an efficient and versatile way. This approach is qualitatively described in Fig. 1g.

**Electrical characterization of MoS₂ FETs.** At first, a commercial silver ink (see Methods) was chosen to print the electrodes, because it can assure very high conductivity with just one printing pass, and it has shown ohmic contact with MoS₂[29]. Typical transfer and output characteristics of the MoS₂ FETs are reported in Fig. 2a, c. The devices work in the enhancement mode, can operate at low supply voltage (<2 V), and exhibit a threshold voltage ($V_{TH}$) in the range of ±1 V (see Supplementary Fig. 7c and

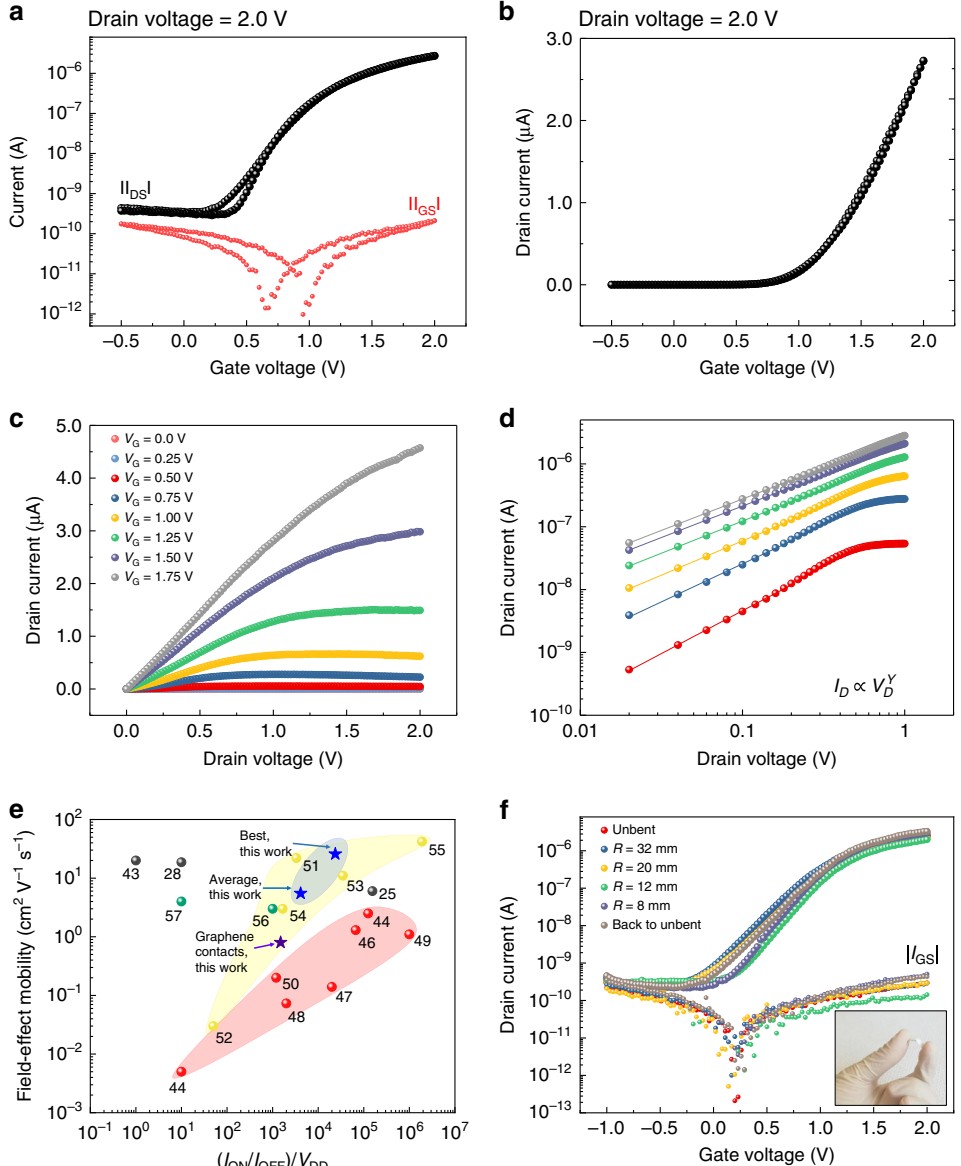

**Fig. 2 Electrical characterization of the MoS$_2$ FETs with inkjet-printed silver contacts in ambient conditions. a** Typical transfer characteristic curve measured as a function of the gate voltage for a drain voltage of 2.0 V. Logarithmic scale: black dots, drain current; red dots, gate current. **b** Typical transfer characteristic curve measured as a function of the gate voltage for a drain voltage of 2.0 V in linear scale. **c** Typical output characteristic measured at different gate voltages (from $V_{GS} = 0.0$ V to $V_{GS} = 1.75$ V, steps of 0.25 V). **d** Log–log curves of the output characteristic in low drain voltage region. Ohmic behavior is observed, suggesting good electrical contact between the silver contacts and MoS$_2$. **e** Field-effect mobility and $(I_{ON}/I_{OFF})/V_{DD}$ for FETs characterized on paper substrates previously reported in the literature. $V_{DD}$ is the supply voltage for each device. Blue stars, this work, inkjet-printed silver contacts; purple star, this work, inkjet-printed graphene contacts; black dots, 2D materials (25, 28, 43); red dots, organic semiconductors (44, 45, 46, 47, 48, 49, 50); yellow dots, inorganic oxides (51, 52, 53, 54, 55); green dots, CNTs (56, 57). **f** Transfer characteristics and gate leakage currents measured for different bending radii along the current direction for a drain voltage of 2 V; inset, picture of a sample with MoS$_2$ FET fabricated on paper.

Supplementary Fig. 7e). As can be seen, the leakage current $I_{GS}$ (red dots, Fig. 2a) through the insulator is negligible as compared to the drain current $I_{DS}$ (black dots, logarithmic scale, Fig. 2a; black dots, linear scale, Fig. 2b), further confirming the good insulating properties of the inkjet-printed hBN film. The saturation regime is reached for low drain-to-source voltage ($V_{DS}$), i.e., $V_{DS} < 2$ V. Almost negligible contact resistance is shown from the output characteristic. Indeed, as can be seen from the log–log plot (Fig. 2d), the linearity parameter $\gamma$, describing the relation $I_{DS} \propto V_{DS}^{\gamma}$, is found to be 1.1 on average, indicating a good contact between the CVD MoS$_2$ and the inkjet-printed silver electrodes.

Charge carrier field-effect mobility ($\mu_{FE}$) is one of the most important figures of merit defining the quality of transistor electrical performance. It can be extrapolated using the classical model for devices operating in the saturation regime ($V_{DS} > V_{GS} - V_{TH}$):

$$\mu_{FE} = 2 \frac{L}{W} \frac{1}{C_i} \left( \frac{\partial \sqrt{I_{DS}}}{\partial V_{GS}} \right)^2 \qquad (1)$$

where $C_i$ is the capacitance of the insulator per unit area, $W$ and $L$ are the transistor channel width and length, respectively, and $V_{GS}$ is the gate voltage. As suggested in ref. [8], in order to avoid any mobility overestimation, the capacitance was measured under

quasi-static conditions (details about the quasi-static capacitance measurement and setup are reported in Supplementary Note 3). To this purpose, parallel plate capacitor structures, in which hBN is sandwiched between silver bottom/top electrodes were fabricated and tested (see Methods). The extracted average value of $230\,nF\,cm^2$ is in line with other quasi-static measurement performed on both organic and hybrid materials[8,36–38]. Thanks to the high capacitive coupling, which results in an enhanced polarization and leads to a high number of carriers at the insulator-semiconductor interface, the devices show an average charge carrier mobility of $5.5\,cm^2\,V^{-1}\,s^{-1}$ and $I_{ON}/I_{OFF}$ ratio of $8 \times 10^3$. $I_{ON}$ is computed for $I_{DS}$ extracted for gate voltage $V_{GS} = V_{GSoff} + V_{DD}$ and drain voltage $V_{DS} = V_{DD}$, where $V_{DD}$ is the supply voltage, and $V_{GSoff}$ is the gate voltage for the lowest current flowing in the device, i.e., the OFF current $I_{OFF}$[39]. The detailed electrical characterization is reported in Supplementary Figs. 2 and 3. Remarkably, this mobility value is comparable to already reported CVD-grown $MoS_2$ transistors fabricated on rigid substrates using standard microelectronic fabrication techniques[40–42], confirming that our methodology, based on the channel array, allows to use inkjet-printing for the fabrication of the devices, without affecting the electronic properties of the channel.

Figure 2e shows $\mu_{FE}$ and the normalized $I_{ON}/I_{OFF}$ ratio for our devices compared with those previously reported in the literature: the closer the points to the top-right corner, the better the performance. We considered only transistors fully fabricated on paper or transferred on paper after fabrication. For a fair comparison, all the $I_{ON}/I_{OFF}$ values are re-calculated considering the International Technology Roadmap for Semiconductors definition[39], and then divided by the respective supply voltage $V_{DD}$. This normalization allows to take into account the operating voltage ranges of the considered FETs, which is a crucial problem for portable applications, where low power consumption is often required. They have been divided into four groups according to the nature of the semiconductor used as channel: 2D materials[25,28,43] organic semiconductors[44–50], inorganic oxide semiconductors[51–55], carbon nanotubes[56,57]. Our devices show competitive electrical performance and are the only one, where both the contacts and the insulating layers are deposited by means of inkjet printing (for a detailed comparison see Supplementary Data 1). While maintaining a high $I_{ON}/I_{OFF}$ ratio, the mobility values extracted from the $MoS_2$ FETs are larger than those obtained for organic semiconductors. Transistors that show comparable or better performance than those presented in this work, as reported in refs. [25,51,53,55], were fabricated using micro-fabrication techniques for the deposition of insulator and contact layers, as well. It is worth mentioning, that the mobility extracted in this work is comparable to the one found for CVD $MoS_2$ FETs entirely fabricated using conventional microelectronic techniques on planarized paper substrates[25]. Note that the type of paper used in the literature may be different, and this may affect the performance of the devices and the reproducibility, hence comparison should be done carefully. In some cases, planarization layers were introduced to mitigate the surface roughness of the paper substrates and high-temperature processes were employed, thus increasing the complexity of the transistor manufacturing. In our work, the fabrication process and the electrical characterization are carried out at ambient condition on a commercially available paper, designed for printed electronics (see Supplementary Note 6) that cannot withstand temperatures over 120 °C. The potentiality of our approach stands in the coherent combination of two large-area fabrication processes in order to obtain good electrical performances. In Supplementary Note 5 and Supplementary Data 2, a comparison with devices fabricated on flexible substrates (other than paper) is reported. As can be seen, our devices are comparable with the best-in-class presented devices.

In order to confirm the compatibility of our technology with flexible substrates, the electromechanical properties of the devices are investigated for various bending radii (R) (more details can be found in Supplementary Note 4.1). Figure 2f shows transfer characteristics recorded for R values of 32, 20, 12, and 8 mm. No relevant changes both in the drain and the gate currents are observed, indicating that the device electrical performance is not affected under the applied strain conditions.

We have then focused on the fabrication of fully 2D-material-based transistors with inkjet-printed graphene source, drain, and gate contacts (see Methods). An optical micrograph of a fully 2D-material-based FET is shown in Fig. 3a, whereas Fig. 3b, c report typical transfer and output characteristic, respectively. These transistors show a reduced effective $\mu_{FE}$ (~0.8 $cm^2\,V^{-1}\,s^{-1}$), and an $I_{ON}/I_{OFF}$ ratio about one order of magnitude smaller (~3 × $10^3$), compared with transistors with silver contacts (Fig. 2e). The reduced performance is likely related to the formation of Schottky contacts, which in turn increases the contact resistance, as evident from Fig. 3c, where non-linear behavior of the output characteristic can be observed for small $V_{DS}$. However, it is remarkable that the extracted field-effect mobility is only six times smaller than the one obtained for a thin film transistor with exfoliated $MoS_2$ channel and CVD-grown graphene source/drain electrodes (4.5 $cm^2\,V^{-1}\,s^{-1}$)[58].

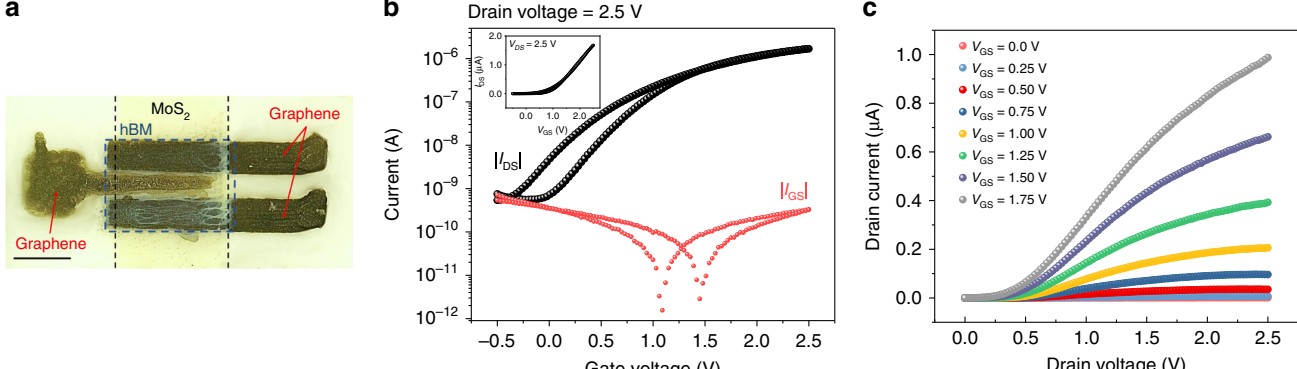

**Fig. 3 Optical image and electrical characterization of a fully 2D-material-based FETs. a** Optical micrograph of a fully 2D-material-based transistor on paper. The scale bar corresponds to 250 μm. **b** Typical transfer characteristic curve measured as a function of the gate voltage for a drain voltage of 2.5 V. Logarithmic scale: black dots, drain current; red dots, gate current. inset, Typical transfer characteristic curve measured as a function of the gate voltage for a drain voltage of 2.5 V in linear scale. **c** Typical output characteristic curves measured at increasing gate voltages (from $V_{GS} = 0.0$ V to $V_{GS} = 1.75$ V, steps of 0.25 V).

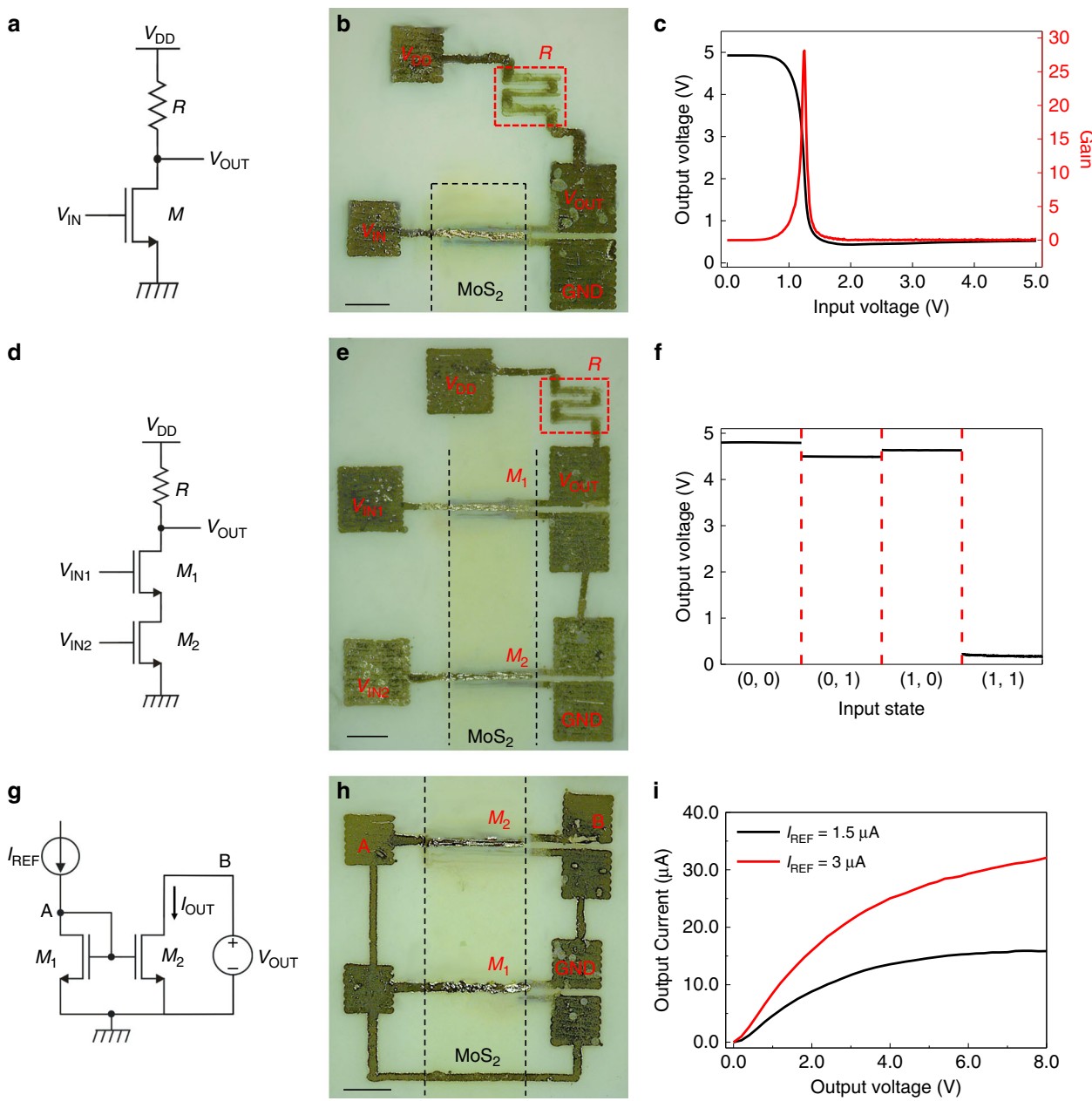

**Fig. 4 Logic gates and current mirror based on inkjet-printed MoS$_2$ FETs. a** Electrical schematic and **b** optical image of an inverter. **c** Output voltage (left axis) and voltage gain (right axis) of the inverter gate as a function of the input voltage. **d** Electrical schematic and **e** optical image of a NAND gate. **f** Output voltage of the NAND gate as a function of the input states ($V_{IN1}$, $V_{IN2}$). Voltage bias is 5 V for both the inverter and the NAND gate. **g** Electrical schematic and **h** optical image of a current mirror. **i** Output current of the current mirror as a function of the output voltage for two different values of the reference current. Legend: $V_{DD}$, supply voltage; GDN, ground reference; $V_{IN}$ and $V_{OUT}$, input and output voltage; $M$, inkjet-printed transistor. $R$, inkjet-printed graphene resistor. The scale bars in **b**, **e**, **h** correspond to 250 μm.

**Integrated circuits on paper**. To demonstrate the potential of the fabricated FETs as building blocks for integrated circuits exploiting the channel array technology, different types of circuits have been designed and fabricated. MoS$_2$ FETs with inkjet-printed silver contacts were selected owing to the high $I_{ON}/I_{OFF}$ ratio, intrinsic gain, and low power supply voltage to fabricate a resistor-transistor logic (RTL) inverter, consisting of a transistor and an inkjet-printed graphene resistor. Figure 4a, b show the schematic and the optical image of a RTL inverter, respectively. The transfer characteristic of an inverter is shown in Fig. 4c (left axis), together with the gain $G$ (right axis), defined as the slope $dV_{OUT}/dV_{IN}$ of the transfer curve (where $V_{IN}$ and $V_{OUT}$ are the input and output voltages, respectively).

The inverter exhibits a high-gain value, close to 30 under a voltage bias of 5 V, in agreement with previously reported inverters based on CVD-grown MoS$_2$ fabricated on rigid substrates[41,59].

The schematic and the optical image of a NAND gate are shown in Fig. 4d, e, whereas Fig. 4f shows the output voltage of the circuit as a function of the inputs ($V_{IN1}$, $V_{IN2}$). The low and high logic values of the inputs correspond to voltages of 0 V and 5 V, respectively. The output voltage is high (i.e., at logic state "1") when at least one input is in the logic state "0", and therefore at least one transistor is in the OFF state. The output voltage is low (i.e., at logic state "0") only when both inputs are at the logic state "1": in these conditions both transistors are in the ON state. The

possibility to implement NAND gates is particularly important, as all other logic functions can be implemented using NAND gates.

As a further demonstration of the potential of the presented technology, we propose an application for analog electronics. Current mirrors are fundamental building blocks in analog electronic circuits, where they are widely used for operational amplifiers, bandgap voltage reference, etc.[60], and they can also be exploited in neural networks, in order to implement matrix-vector multiplication[61]. Figure 4g, h show the schematic and the optical image of the fabricated current mirror. The output transistor ($M_2$) is 10 times wider than the input transistor ($M_1$), whereas all the other transistor parameters are identical; therefore, the current mirror has a nominal gain of 10. Figure 4i shows the current mirror output characteristic, i.e., the output current as a function of the output voltage, for two different values of the input reference current. As shown in the plot, for sufficiently high output voltages (i.e., $M_2$ in saturation) the output current is about 10 times larger than the reference current ($I_{REF}$), in accordance with the circuit design.

## Discussion

We have successfully demonstrated high-performance $MoS_2$-based transistors that combine the numerous advantages of using paper as a substrate with the versatility of inkjet-printing technique, whilst maintaining the good electrical properties of CVD-grown $MoS_2$. A maximum field-effect mobility of 26 cm$^2$ V$^{-1}$ s$^{-1}$ and an $I_{ON}/I_{OFF}$ ratio of up to $5 \times 10^4$ were achieved. Bending tests have shown that the device electrical properties are robust under applied strain (up to a bending radius of 8 mm). Moreover, our device fabrication approach has been proven to be suitable for the development of complete integrated circuits, such as high-gain inverters, logic gates, and current mirrors. This work demonstrates the great potential of the channel array technology for next-generation electronics on paper, ranging from analogic to digital circuits for cost-efficient and practical applications.

## Methods

**Materials**. PEL P60 (purchased from Printed Electronics Limited) is used as paper substrate (more details can be found in Supplementary Note 6). A commercial silver ink (Sigma-Aldrich) is used to print the metal contacts. Bulk graphite (purchased from Graphexel or Sigma-Aldrich, 99.5% grade) and bulk boron nitride (purchased from Sigma-Aldrich, >1 μm, 98% grade) powders were used to prepare the 2DMs inks. The bulk powders are dispersed in deionized water (resistivity 18.2 MΩ cm$^{-1}$) at a concentration of 3 mg mL$^{-1}$ and 1-pyrenesulphonic acid sodium salt (PS1, purchased from Sigma-Aldrich), purity ≥97%, is added at a concentration of 1 mg mL$^{-1}$. The graphite and boron nitride dispersions are then sonicated for 72 h and 120 h, respectively, using a 300 W Hilsonic HS 1900/Hilsonic FMG 600 bath sonicator at 20 °C. The resultant dispersions is centrifuged at 3500 rpm ($g$ factor = 903) for 20 minutes at 20 °C using a Sigma 1–14 K refrigerated centrifuge in order to separate out and discard the residual bulk, non-exfoliated flakes. The remaining supernatant, now containing the correct flake size and monolayer percentage, is centrifuged twice to remove excess PS1 from the dispersion. After washing, the precipitate is re-dispersed in the printing solvent, made as described in ref. [16]. The concentration of the resultant inks are assessed using a Varian Cary 5000 UV-Vis spectrometer and the Lambert-Beer law, with extinction coefficients of 2460 (at 660 nm) and 1000 L g$^{-1}$ m$^{-1}$ (at 550 nm) for graphene[11] and hBN[62], respectively. Full characterization of the material (lateral size and thickness distribution, crystallinity, etc.) has been presented in refs. [16,34,63].

**Growth of MoS$_2$ and transfer on paper**. Single- and few-layer $MoS_2$ have been grown by CVD on c-plane sapphire[64]. CVD growth is performed at atmospheric pressure using ultra-high-purity Ar as the carrier gas. The substrates are placed face-down 15 mm above a crucible containing ~3 mg of MoO$_3$ (99.998% Alfa Aesar) and loaded into a split-tube two-zone CVD furnace with a 30 mm outer diameter quartz tube. A second crucible containing 1 g of sulfur (99.9% purity, Sigma-Aldrich) is loaded upstream from the growth substrates and heated to 140 °C. Just before starting the growth, the tube is flushed with Ar at room temperature and atmospheric pressure. The furnace is pre-heated to 750 °C for few hours for temperature stabilization. The substrate and MoO$_3$ precursor were then loaded into the growth area of the furnace to start the growth. Ar is supplied with a flow of 10 sscm. After 10 min growth at 750 °C, the furnace is left to cool naturally. The synthesized $MoS_2$ film is transferred from the native sapphire substrate to the paper substrates[65]. Prior to the film transfer, $MoS_2$ is patterned to obtain a stripped

structure on the growth substrate. The pattern is defined by means of Ar/SF$_6$ plasma etching in an Oxford Cobra Reactive Ion Etching system. The etch mask is created by optical lithography using AZ 5214E photoresist. A 5% KOH solution in DI water is employed to remove the etch mask. The sample is then rinsed in DI water for several times to remove the KOH remnants. To transfer the patterned film, the sapphire substrate with the $MoS_2$ layer film is first covered with a poly-styrene film by spin coating a solution of polystyrene in toluene onto the substrate, which is then immersed in DI water. To facilitate the lift-off process, a solution of KOH in DI water is also added for a short time. After that, the carrier polystyrene film with the $MoS_2$ layer is rinsed for several times in DI water. To remove the absorbed water the polymeric film is dried at 50 °C in dry air atmosphere and then transferred onto the paper substrate. To improve the adhesion between the carrier polymer film and the wafer the sample is baked at 150 °C for about an hour. The polystyrene carrier film is then dissolved in toluene resulting in a $MoS_2$ film on paper. Raman characterization before and after transfer has been performed with a Renishaw InVia spectrometer equipped with a confocal optical microscope and a 532 nm excitation laser. The spectral resolution of the system is 1 cm$^{-1}$. Raman experiments were carried out employing a ×50 objective (N.A. 0.6), laser power of 5 mW and an acquisition time of 2 s. The pixel size is 1 μm × 1 μm.

**Devices fabrication**. $MoS_2$ transistors are fabricated in a top-gate/top-contact configuration on the CVD $MoS_2$ stripes transferred on paper. A Dimatix Materials Printer 2850 (Fujifilm) is used to define the contacts and the insulator layers under ambient conditions. It is worth underlining that no annealing or post-treatment process is performed after any printing step. The silver ink is deposited with a single printing pass using one nozzle, a drop spacing of 40 μm, and keeping the printer platen at room temperature. Cartridges with a typical droplet volume of 1 pL are used for the definition of the contacts. When the 2.5 mg ml$^{-1}$ graphene ink is employed, source and drain contacts are inkjet-printed using a drop spacing of 20 μm, and 20 printing passes. Cartridges with a droplet volume of 10 pL are used for the definition of the graphene contacts. For the top-gate contacts, only six printing passes of graphene ink at the same concentration are used in order to reduce the possibility of overlapping with the source and drain contacts (which would significantly increase the leakage current) at each print pass. A ~2 mg/mL hBN ink is printed on top of the CVD-grown $MoS_2$ using a drop spacing of 20 μm and 80 printing passes. Cartridges with a droplet volume of 10 pL are used for the definition of the insulating layer. Several transistors have been fabricated (with a yield of around 80%) and characterized with a nominal width of ~500 μm and length varying between 40 μm and 60 μm (further details can be found in Supplementary Table 1).

Parallel plate capacitors with silver/graphene bottom/top electrodes are also printed on paper to evaluate the capacitance of the hBN layers. The inks and the fabrication procedures are kept the same for all the reported devices.

**Electrical characterization**. All the electrical measurements are performed under ambient conditions. The transistor characterization is carried out using a Keithley SCS4200 parameter analyzer. Capacitance measurements are performed with an R&SRTO2014 oscilloscope and a HP 33120A function/arbitrary waveform generator. The detailed description of the measurement setup can be found in the Supplementary Information.

## Data availability

The data that support the findings of this work are available from the corresponding authors upon reasonable request.

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

## Acknowledgements

We acknowledge the ERC PEP2D (contract no. 770047) and H2020 WASP (contract no. 825213) for financial support. C.Ca., C.Co., G.F.,T.M. acknowledge the Graphene Flagship Core 3 (contract no. 881603). R.W. acknowledges the Hewlett-Packard Company for financial support in the framework of the Graphene NOWNANO Centre for Doctoral Training;. C.Ca. acknowledges useful discussions with Alessandro Molle, and financial support from the Grand Challenge EPSRC grant EP/N010345/1. S.P. and C.Co acknowledge financial support from Compagnia di San Paolo (project STRATOS).

## Author contributions

R.W., S.M. developed the inks under the supervision of C.Ca.; D.K.P., M.P., S.P., D.H.K., and F.F. carried out the $MoS_2$ growth and the transfer on the paper substrates, and performed the Raman spectroscopy and imaging under the supervision of T.M and C.Co.; S.C., L.P., G.C., and G.F. fabricated the electronic devices, performed the electrical measurements, and analyzed the results; G.F, G.I., and M.M. designed and supervised the research. All authors discussed the results and contributed to the manuscript.

## Competing interests

The authors declare no competing interests.

**Additional information**

