## [Peer Review File · Nature Communications]

Reviewers' comments:

Reviewer #1 (Remarks to the Author):

The manuscript describes the use of chemical vapor deposition MoS₂ with inkjet printing of a h-BN dielectric and silver electrodes to fabricate flexible MoS₂ field-effect transistors on paper with a current on/off ratio 10⁵ and mobility of 15 cm² V⁻¹ S⁻¹. The MoS₂ transistors are then combined with printed graphene resistors and silver interconnects to create inverters, logic gates and current mirrors.

The technique proposed by the authors to use chemical vapor deposition MoS₂ as a channel material with inkjet printed components is not new. For example, Kim, et al. ACS Nano 2016, 10, 2, 2819-2826 (doi.org/10.1021/acsnano.5b07942) has demonstrated CVD grown MoS₂ with inkjet printed silver contacts achieving a mobility of 1.8 cm²/V s and on/off ratio 10⁴ while Kim, et al. ACS Nano 2017, 11, 10, 10273-10280 (doi.org/10.1021/acsnano.7b04893) has also used CVD grown MoS₂ with inkjet printed PVP and PEDOT to build a transistor on a flexible substrate (PEN) achieving a mobility of 0.27 cm²/V s and on/off ratio of 10³. The combination of CVD material with solution processed material in a heterostructure has already been published by the authors (Withers, F. et al. Nano Letters 14, 3987-3992 (2014). DOI: 10.1021/nl501355j). For this reason, in my opinion, the manuscript is incremental work and would not be suitable for Nature Communications.

On page 8 the authors say "references 18,40, 42, 44, that show comparable or better performance than ours, we remark that, in those cases, expensive, and time-consuming fabrication techniques were employed for their fabrication." However this is incorrect as the manuscript uses CVD MoS₂ which is an expensive and time consuming technique. The CVD MoS₂ is also transferred onto paper using a conventional wet-transfer method which is just as time-consuming as the wet transfer technique in reference 18. For example, reference 18 uses PMMA-assisted wet transfer of CVD grown MoS₂ onto a paper substrate and then also achieves a better average mobility 6 cm²/ V s and on/off ratio 10⁹ than this manuscript.

On page 4 the authors state "an average field effect mobility of 5.9 cm²/ V s" "and Ion/Ioff ratios up to 10⁴, which have never been recorded on paper". This is also incorrect, as reference 18 also uses CVD MoS₂ on a paper substrates and achieves better mobility and on/off ratio.

On page 8 the authors state "the devices show an average charge carrier mobility of 5.9 cm²/ V s and ION/IOFF ratio of 1.4x10⁴" however this on/off ratio doesn't agree with what is stated in the abstract and conclusion. The abstract states "current modulation (Ion/Ioff up to 10⁵) and mobility (up to 15 cm² V⁻¹ s⁻¹)". So what is the performance of the device? Please discuss.

On page 12 the authors describe an n-type inverter with a gain ~30. Two of the authors have already demonstrated n-type inverters with CVD MoS₂ which are double this value ~60 Wachter, S et. al. Nature Communications 8, (2017). (doi.org/10.1038/ncomms14948). Lin et al (Reference 19) was able to print n-type inverters with MoS₂ ink which despite being of lower electrical quality than CVD MoS₂ was still able to achieve a gain of a similar magnitude ~20 on a flexible substrate (Kapton).

The authors then describe semi-printed printed low voltage logic gates however low voltage logic gates have already been done fully printed with 2D materials on a flexible substrate (PET) in the literature (Reference 9, Carey et. al).

In the caption of figure 2e the authors compare the "field-effect mobility and Ion/Ioff ratio for TFTs on flexible substrates reported in the literature" however the authors have not shown the results for CVD MoS₂ material on flexible substrates other than paper such as PET, PEN and Kapton. The devices should be compared to state-of-the-art CVD 2D material transistors on flexible substrates.

For example, Gong et. al. 2D Materials 3(2):021008 2016 (doi:10.1088/2053-1583/3/2/021008) has shown CVD WS₂ TFT's on flexible polyimide with on/off ratios of 10⁶ and mobilities of 10 cm²/ V s. Choi et. al. Science Advances, Vol. 4, no. 4, 2018 (DOI: 10.1126/sciadv.aas8721) has shown MoS₂ transistors on PET with a mobility of 20 cm²/ V s and on/off ratio 10⁶. The authors have also already published fully printed devices with carbon nanotubes, h-BN and silver on a flexible Kapton substrate with mobility 10.7 cm²/ V s and on/off ratio 10⁵ (Lu et. al. ACS Nano 2019, 13, 10, 11263-11272, doi: 10.1021/acsnano.9b04337). These papers and many more demonstrate greater values for transistor mobility and on/off ratio on flexible substrates.

On page 9 the authors undertake bending tests to measure the flexibility of their devices, however there is no mention in the draft of the experimental set up making it difficult to reproduce. A description should be added to the methods section. Moreover the thickness of the substrate used should be described so that the reader can calculate the strain that is applied to the device.

There are also typo's in some the reference such as reference 5, 13, 18 and 51

In my opinion, the manuscripts rational is flawed. The authors use an expensive process (CVD) with many fabrication steps with a cheap, fast and scalable technique (inkjet printing). The resulting devices are now expensive to manufacture, with lower figures of merit (mobility and on/off ratio) compared to CVD MoS₂ devices on flexible substrates and also have additional fabrication steps making the technology difficult to scale. The figures of merit presented for mobility and on/off ratio are unremarkable on a flexible substrate.

I recommend to reject this paper on the basis of novelty and unremarkable performance. Scientifically the paper is correct and would therefore recommend publication in a lower impact journal after revision.

Reviewer #2 (Remarks to the Author):

The presented work is very interesting and the performance of the devices is really primising. The device fabrication and characterization has been done properly and with a lot of expertise. The manufacturing approach is novel and smart and is of interest to others in the community and the wider research field.

I would recommend to add a view more general consideration and technical details to the manuscript as listed below.

I have following recommendations:

1) I would recommend to highlight the manufacturing approach and in detail the combination of both manufacturing steps/methods (CVD + inkjet) with their respective technical advantages and disadvantages - also in terms of the potential future industrial use of these technologies for the manufacturing of such devices and the need? of patterning the semiconductor.

I also recommend to highlight the quite smart approach of customizing the transistors by inkjet printing which allows a mass production of not customized/potentially not patterned semiconducting channel layers followed by customizing the transistors using inkjet printing.

2) Related to this statement: "However, paper is a challenging substrate for electronics due to its high roughness and limiting processing temperature and the lack of a winning fabrication technique is preventing its exploitation at the industrial level." I think it is important to add another 1-3 sentences with literature sources (e.g. <https://onlinelibrary.wiley.com/doi/abs/10.1002/adma.201801588>) to this statement in order to indicate some additional challenges

of the concept of paper electronics. Next to roughness and temperature stability, I think it is important to give some indication about further potential limitations/challenges, e.g. about dimensional stability and durability (e.g. as a function of humidity, temperature etc.), functionalization/priming/coatings of paper surfaces, and about reproducibility of paper and paper surface properties (since paper is a natural material).

In addition, I recommend to add more technical details (surface roughness, material composition, ...) about the used paper from PEL which seems to be a very special paper with a special microporous top coat?

3) Please provide a short comparison performance wise to printed transistors on polymer films - either based on your studies (maybe you have also tried to use polymer films instead of paper?) or to literature studies.

4) Please introduce the dielectric Hexagonal Boron Nitride so that the abbreviation hBN becomes more clear.

5) When using silver or graphene inks for the S-D and G electrodes: What do you think about the influence of work function? What was the post-processing (e.g. annealing)?

6) TFT vs FET - I recommend to go for one term only in order to have a more focused terminology.

7) Could you please confirm that 1 pL and not 10 pL (nominal volumes) printheads were used for the deposition of the inks?

8) Please report further details about the printed and CVD processes layer properties, especially (i) thickness/roughness of the dielectric layer after deposition of the layer in so many passes, (ii) W and L of the transistors for all the results/graphs, parameters of the CVD layer.

9) Could you please give more information about the manufacturing yield, some statistics (e.g. do you see any dependencies on W or L, what is the reason for 80% yield, which process is the most challenging process, ...), causes of the transistor performance variability reported in the appendix and impact on the performance of circuits (e.g. taking in account literature such as <https://www.sciencedirect.com/science/article/abs/pii/S1566119914000287>) and your opinion about further industrialization of the processes (e.g. taking in account literature studies of transistors completely manufactured by inkjet printing such as <https://www.nature.com/articles/srep33490>).

10) Please provide some details about the "printability/jettability" of the inks. Is the printing performance stable? Could you apply the inks to other, more productive printheads?

11) Did you see any problems with the different interfaces during the liquid processing (printing), e.g. dewetting effects?

Reviewer #3 (Remarks to the Author):

The manuscript is interesting and reports advances of 2d devices on paper platform. The work shows both device and circuit level integration. While I recognize the advances, the main issues I find are summarized below

-paper is a low-cost substrate ...however, the technique of integration is quite expensive. Is this really an attractive option compared to fully printed or solution based methods of realizing paper

electronics?

- the circuits are non-cmos ...I seriously doubt that RTL logic will be suitable in this modern era. Can the authors demonstrate cmos logic by using additional TMDs for p-type?
- the authors should avoid using such superlative subjective terms like 'excellent'.

Overall I think it is important to address these fundamental questions above.

We thank the Reviewers for their comments, which have certainly allowed us to improve the quality of our work. Here point-by-point answers to their questions follow. Changes in the text have been highlighted in bold.

Reviewer #1 (Remarks to the Author):

R#1_Q1: The technique proposed by the authors to use chemical vapor deposition MoS₂ as a channel material with inkjet printed components is not new. For example, Kim, et al. ACS Nano 2016, 10, 2, 2819-2826 (doi.org/10.1021/acsnano.5b07942) has demonstrated CVD grown MoS₂ with inkjet printed silver contacts achieving a mobility of 1.8 cm²/V s and on/off ratio 10⁴ while Kim, et al. ACS Nano 2017, 11, 10, 10273-10280 (doi.org/10.1021/acsnano.7b04893) has also used CVD grown MoS₂ with inkjet printed PVP and PEDOT to build a transistor on a flexible substrate (PEN) achieving a mobility of 0.27 cm²/V s and on/off ratio of 10³. The combination of CVD material with solution processed material in a heterostructure has already been published by the authors (Withers, F. et al. Nano Letters 14, 3987–3992 (2014). DOI: 10.1021/nl501355j). For this reason, in my opinion, the manuscript is incremental work and would not be suitable for Nature Communications.

R#1_A1: We thank the Reviewer for his/her valuable comments, which have given us the opportunity to better highlight the degree of novelty of our work, that lies in:

- i. the fabrication of **printed Field Effect Transistors on paper, working with low supply voltages (smaller than 2 V), and with notable electrical performance (low threshold voltage and large mobility)**.
- ii. the choice of pre-patterned CVD MoS₂ as semiconducting layers deposited on paper, which gives the possibility to design and fabricate “on-demand” devices and circuits through a mask-less fabrication technique such as inkjet printing. Here, in particular, **we propose an ASIC design approach that we define “channel array”**, echoing the well-known “gate array” approach in the Electrical Engineering community, where only the channel is pre-defined, and all the other elements such as contacts, dielectric, and connections represent a degree of freedom for the designer. **This has never been proposed before (especially on paper) and clearly is far from being incremental.**

As underlined by the Reviewer, the combination of inkjet-printing and CVD for the fabrication of MoS₂ based FETs has already been reported in the literature. In our work, however, we provide a clear demonstration of the feasibility of this technology for the fabrication of the complete device, where, apart from the customized MoS₂ channel array patterns, **all the other components are deposited using inkjet printing**. This allows to fabricate without masks, at ambient conditions, with no annealing steps, fundamental prerequisites to deal with paper, which is our target substrate. Despite this, **the device shows performance comparable (and in some cases even better) to that reported in state-of-the-art**

with conventional fabrication techniques and on rigid substrates. Furthermore, this technology allows to easily build integrated circuits with two-dimensional materials and on paper, which has never been reported so far.

Regarding the previous works cited by the Reviewer, in Kim *et al.* in 2016 (Kim, T.-Y. *et al.* Electrical Properties of Synthesized Large-Area MoS₂ Field-Effect Transistors Fabricated with Inkjet-Printed Contacts. *ACS Nano* **10**, 2819–2826 (2016)) the device was deposited on silicon dioxide, used as dielectric. In our work, where the insulator is made with an inkjet-printed 2D material, the devices show superior performance (*i.e.*, larger mobilities, smaller supply voltages, and higher I_{ON}/I_{OFF} ratios) compared to the work published by Kim *et al.* (Kim, T. Y. *et al.* Transparent Large-Area MoS₂ Phototransistors with Inkjet-Printed Components on Flexible Platforms. *ACS Nano* **11**, 10273–10280 (2017)) where the device was fabricated on plastic substrate (mobility equal to $0.37 \text{ cm}^2 \text{ V}^{-1} \text{ s}^{-1}$ and I_{ON}/I_{OFF} ratio around 10^2 were obtained, while applying large supply voltages).

Concerning Withers' work (Withers, F. *et al.* Heterostructures Produced from Nanosheet-Based Inks. *Nano Lett.* **14**, 3987–3992 (2014)), which has been published by one of the co-authors, it does not report inkjet-printing as well as the use of CVD (this is shown in Figure 1 as a possible substrate, but it was not used in the work), and does not focus on transistors, hence, it is not relevant for this work.

We thank the Reviewer for suggesting these references, which have now been cited in the revised manuscript. The text has been revised to highlight the novelty of our work compared to the state-of-the-art, as follows.

Changes made to the manuscript

- The novelty of the proposed approach has been underlined in the revised version of the introduction: Page 3: **“We combine chemical vapour deposition (CVD), for the growth of high-quality MoS₂ channels, with inkjet-printing, which gives the possibility to design and fabricate customizable devices and circuits exploiting 2DMs-based inks, whose capability to be printed on top of CVD grown materials has been successfully demonstrated in ¹⁶. In this way, an application specific integrated circuit (ASIC) design approach, known as “channel array”, is proposed. It is based on the transfer of strips of CVD-grown MoS₂, the transistor channels, onto a paper substrate where the rest of the devices and circuits, source and drain contacts (defining the channel length and width), gate dielectric, gate contacts, and connections, are fully customized exploiting the inkjet-printing technique, giving a further degree of freedom to the designer. This method allows to keep the flexibility and versatility of an all-inkjet technology, with the difference that here a high-quality channel is already placed on the substrate, by taking advantage of the CVD grown TMDC. Moreover, both being bottom up, large-area fabrication processes, their combination could open a possible exploitation at the industrial level.”**

~~We propose a new fabrication technique that combine the use of high mobility MoS₂ channels grown by CVD and inkjet printing technique for the fabrication of transistors and circuits, exploiting 2DMs based inks, whose capability to be printed on top of CVD grown materials has been successfully demonstrated in¹⁶. This approach is called “channel array”, as it is based on the deposition of several channels on a customized substrate, onto which the rest of the device and circuit is fabricated. The “channel array” technique consists of two steps: first, the channel array is fabricated by transferring on the paper substrate stripes of CVD grown MoS₂, which will be used as transistors channels. In the second step, the FETs and the rest of the circuit are fabricated by customizing the channel array by inkjet printing the source and drain contacts (i.e., channel length and width), gate dielectric, gate contacts, and all connections. This approach allows to keep the flexibility and versatility of an all inkjet technology, with the difference that here a high quality channel is already placed on the substrate, by taking advantage of the CVD grown TMDC.~~

- For a better comparison with the state-of-the-art, we expanded *Section S5 “Comparison with the literature”* of the supplementary information, adding one comparative graph, and the relative literature table (see Figure S8, Table S2, and R#1_A7).

R#1_Q2: On page 8 the authors say "references 18, 40, 42, 44, that show comparable or better performance than ours, we remark that, in those cases, expensive, and time-consuming fabrication techniques were employed for their fabrication." However, this is incorrect as the manuscript uses CVD MoS₂ which is an expensive and time consuming technique. The CVD MoS₂ is also transferred onto paper using a conventional wet-transfer method which is just a time-consuming as the wet transfer technique in reference 18. For example, reference 18 uses PMMA-assisted wet transfer of CVD grown MoS₂ onto a paper substrate and then also achieves a better average mobility 6 cm²/ V s and on/off ratio 10⁹ than this manuscript.

R#1_A2: We thank the Reviewer for his/her comments. As reported in the previous answer, we have now added a more comprehensive discussion about the state-of-the-art of flexible MoS₂-based FETs and the sentence mentioned by the Reviewer has been removed, since indeed, it could have led to possible misunderstandings. Regarding the mentioned performance in Ref. 18 (Park, S. & Akinwande, D. First demonstration of high performance 2D monolayer transistors on paper substrates. in 2017 IEEE International Electron Devices Meeting (IEDM) 5.2.1-5.2.4 (IEEE, 2017), now Ref. 25), we want to highlight that, in our case, the obtained field effect mobility 5.5 cm² V⁻¹ s⁻¹ is an average mobility obtained over 26 devices, with a maximum mobility of 26 cm² V⁻¹ s⁻¹, well comparable with that reported in Ref. 18 (now Ref. 25). This is very exciting, considering that every part of the device, apart from the channel, has been fabricated using a “dirty” technique such as inkjet-printing, in contrast to the device reported in Ref. 18 (now Ref. 25), which was made with clean-room based technologies and post-processing, such as annealing (not compatible with our paper substrate that cannot withstand temperatures higher than 120°C).

Regarding instead the I_{ON}/I_{OFF} , we thank the Reviewer for pointing this out, since it gives us the possibility to clarify and improve the comparison.

Indeed, we would like to underline that the values for the I_{ON}/I_{OFF} ratios available in the literature have often been extracted with arbitrary definitions of the ratio. In order to make a fair comparison, we have then recalculated and compared the values presented in the literature using what we believe is the most rigorous definition, i.e., the one available in the ITRS (<http://www.itrs2.net/>): I_{ON} is computed for I_{DS} (drain-to-source current) extracted for gate voltage $V_{GS} = V_{GSoff} + V_{DD}$ and drain voltage $V_{DS} = V_{DD}$, where V_{DD} is the supply voltage and V_{GSoff} is the gate voltage for the lowest current flowing in the device, i.e., the OFF current I_{OFF} . An average I_{ON}/I_{OFF} close to 10^4 was obtained for our devices. Following this definition, and for a fair comparison, the I_{ON}/I_{OFF} ratio reported in Ref. 18 (now Ref. 25) is, in the best-case scenario, of the order of 10^5 .

Changes made to the manuscript

- The I_{ON}/I_{OFF} values of our devices have all been recalculated following the ITRS definition. The new average value has been reported throughout the text. The statistic in the supplementary information has been modified and shown in Figure S7.

- The main text has been changed as follow:

Page 10: “Thanks to the high capacitive coupling, which results in an enhanced polarization and leads to a high number of carriers at the insulator-semiconductor interface, the devices show an average charge carrier mobility of $5.5 \text{ cm}^2 \text{ V}^{-1} \text{ s}^{-1}$ and an I_{ON}/I_{OFF} ratio of 8×10^3 is obtained. I_{ON} is computed for I_{DS} extracted for gate voltage $V_{GS} = V_{GSoff} + V_{DD}$ and drain voltage $V_{DS} = V_{DD}$, where V_{DD} is the supply voltage, and V_{GSoff} is the gate voltage for the lowest current flowing in the device, i.e., the OFF current I_{OFF} ³⁸. ~~where I_{ON} is defined as the I_{DS} measured when V_{DS} is equal to V_{GS} ($V_{GS} = V_{DS} = 2\text{V}$), while I_{OFF} is the average I_{DS} in the subthreshold region ($V_{GS} < V_{TH}$).~~

Page 11: “Transistors that show comparable or better performance than ours, as reported in references^{25,50,52,54}, were fabricated using micro-fabrication techniques for the deposition of insulator and contact layers, as well.”

~~Although there are transistors, as reported in references 18,40,42,44, that show comparable or better performance than ours, we remark that, in those cases, expensive, and time-consuming fabrication techniques were employed for their fabrication.~~

R#1_Q3: On page 4 the authors state “an average field effect mobility of $5.9 \text{ cm}^2 / \text{V s}$ ” and I_{on}/I_{off} ratios up to 10^4 , which have never been recorded on paper”. This is also incorrect, as reference 18 also uses CVD MoS_2 on a paper substrates and achieves better mobility and on/off ratio.

R#1_A3: Regarding the performance and comparison with Ref. 18 (now Ref. 25), please refer to the previous question. We recognize the inaccuracy of our statement, which has been modified.

Changes made to the manuscript

- We modified the main text as follows:

Page 5: “ The MoS₂ field effect transistors fabricated with the channel array method operate at supply voltage below 2 V, with remarkable transistor performance, such as an average field effect mobility of **5.5 cm² V⁻¹ s⁻¹ (with best performance reaching 26 cm² V⁻¹ s⁻¹), negligible leakage currents (smaller than 5 nA), and an average I_{ON}/I_{OFF} ratio of 8×10^3 (up to 5×10^4).** ~~The MoS₂ field-effect transistors fabricated with the channel array method operate at supply voltage below 2 V, with remarkable transistor performance, such as an average field effect mobility of 5.9 cm² V⁻¹ s⁻¹ (up to 15 cm² V⁻¹ s⁻¹), negligible leakage currents (smaller than 1 nA), and I_{ON}/I_{OFF} ratios of up to 10⁵, which have never been recorded on paper.~~

R#1_Q4: On page 8 the authors state "the devices show an average charge carrier mobility of 5.9 cm²/V s and I_{ON}/I_{OFF} ratio of 1.4×10^4 " however this on/off ratio doesn't agree with what is stated in the abstract and conclusion. The abstract states "current modulation (I_{ON}/I_{OFF} up to 10^5) and mobility (up to 15 cm² V⁻¹ s⁻¹)". So what is the performance of the device? Please discuss.

R#1_A4: Thanks for pointing this out. Actually, in the abstract we were mentioning the best performance, as often done in the literature: this has been clearly stated in the revised version of the manuscript. Triggered by the Reviewer comments, we have doubled the number of devices that are included in the statistics. An average field effect mobility of 5.5 cm² V⁻¹ s⁻¹ and an average I_{ON}/I_{OFF} of 8×10^3 are obtained, as reported in Figure S7.

Changes made to the manuscript

- We doubled the number of fabricated devices. The statistic in the supplementary information has been modified and shown in Fig S7.
- The abstract has been modified as:
"with an average current modulation I_{ON}/I_{OFF} of 8×10^3 and mobility of 5.5 cm² V⁻¹ s⁻¹, which in the best case reaches values of 5×10^4 and 26 cm² V⁻¹ s⁻¹, respectively" ~~with excellent current modulation (I_{ON}/I_{OFF} up to 10^5) and mobility (up to 15 cm²/Vs).~~
- The updated mobility and I_{ON}/I_{OFF} values have been reported in the conclusions:
Page 17: “A maximum field-effect mobility of **26 cm² V⁻¹ s⁻¹** and an I_{ON}/I_{OFF} ratio of up to **5×10^4** were achieved.”

R#1_Q5: On page 12 the authors describe an n-type inverter with a gain ~30. Two of the authors have already demonstrated n-type inverters with CVD MoS₂ which are double this value ~60 Wachter, S et. al. Nature Communications 8, (2017). (doi.org/10.1038/ncomms14948). Lin et al (Reference 19) was able to print n-type inverters with MoS₂ ink which despite being of lower electrical quality than CVD MoS₂ was still able to achieve a gain of a similar magnitude ~20 on a flexible substrate (Kapton).

R#1_A5: We thank the Reviewer for the valuable comments, although we think that they may come from a misunderstanding about the novelty of our work. Our work does not show conceptually new devices, as also stated by the Reviewer, but a new way to fabricate devices that does not lower their performance, when compared to devices fabricated through complex micro-fabrication techniques. As underlined in the main manuscript (page 3), Lin *et al.* (Lin, Z. *et al.* Solution-processable 2D semiconductors for high-performance large-area electronics. *Nature* **562**, 254–258 (2018)), Ref. 19, now Ref. 27) reported transistors fabricated with solution-processed MoS₂, which show remarkable performance (average mobility of around 7–11 cm² V⁻¹ s⁻¹). However, the MoS₂ solution was deposited by spin coating and the electrical characterization presented is relative to the devices fabricated on rigid SiO₂/Si substrate (as shown in *Extended Data Fig. 8* of the manuscript), and not on Kapton. In addition, the fabrication steps require acid cleaning and annealing above 300 °C, which are incompatible with substrates such as paper, which is our target substrate.

Likewise, the work presented by Wachter *et al.* (old Ref. 32, now Ref. 40; Wachter, S., Polyushkin, D. K., Bethge, O. & Mueller, T. A microprocessor based on a two-dimensional semiconductor. *Nat. Commun.* **8**, 14948 (2017)) showed excellent results for CVD MoS₂ on rigid silicon substrates, with both metal contacts and dielectric layers deposited using conventional microfabrication techniques. As a validation of the good quality of this active layer, in our work, a gain of around 30 was obtained on paper substrate with inkjet-printed contacts and dielectric layers. ***We want to stress that achieving results comparable, or even better than those obtained on rigid substrate with micro-fabrication techniques, on paper, is very challenging and this clearly represents one of the biggest achievements of our work.***

R#1_Q6: The authors then describe semi-printed low voltage logic gates, however low voltage logic gates have already been done fully printed with 2D materials on a flexible substrate (PET) in the literature (Reference 9, Carey et. al).

R#1_A6: We thank the Reviewer for pointing out this work, which indeed has been cited in the text. We however, do not think that our work can be compared to Ref. 9 (Carey, T. *et al.* Fully inkjet-printed two-dimensional material field-effect heterojunctions for wearable and textile electronics. *Nat. Commun.* **8**, 1202 (2017), now Ref. 15) results. In Ref. 9 (now Ref. 15), an attempt to explore graphene FETs for logic application has been done. However, as well-known, graphene does not have a bandgap, which prevents its use for logic applications. As indeed can be seen in Ref. 9 (now Ref. 15), the obtained gain of the inverter is very small, i.e. of the order of 0.1, while, for logic level regeneration, needs to be at least one. Moreover, as also stated by the authors, the output swing is also small.

As we believe it is clear from the work we presented, our results are way beyond those obtained with graphene technology.

R#1_Q7: In the caption of figure 2e the authors compare the "field-effect mobility and Ion/Ioff ratio for TFTs on flexible substrates reported in the literature" however the authors have not shown the results

for CVD MoS₂ material on flexible substrates other than paper such as PET, PEN and Kapton. The devices should be compared to state-of-the-art CVD 2D material transistors on flexible substrates. For example, Gong *et al.* *2D Materials* 3(2):021008 2016 (doi:10.1088/2053-1583/3/2/021008) has shown CVD WS₂ TFT's on flexible polyimide with on/off ratios of 10⁶ and mobilities of 10 cm²/V s. Choi *et al.* *Science Advances*, Vol. 4, no. 4, 2018 (DOI: 10.1126/sciadv.aas8721) has shown MoS₂ transistors on PET with a mobility of 20 cm²/V s and on/off ratio 10⁶. The authors have also already published fully printed devices with carbon nanotubes, h-BN and silver on a flexible Kapton substrate with mobility 10.7 cm²/V s and on/off ratio 10⁵ (Lu *et al.* *ACS Nano* 2019, 13, 10, 11263-11272, doi: 10.1021/acsnano.9b04337). These papers and many more demonstrate greater values for transistor mobility and on/off ratio on flexible substrates.

R#1_A7: We thank the Reviewer for his/her comments. Actually, as stated above, we want to underline again that the values for the I_{ON}/I_{OFF} ratios reported in literature cannot be directly compared, since most of the time they have been extracted with arbitrary definitions of the ratio (See R#1_A2). For a fair comparison, the values in the literature have been extracted again following the ITRS definition. To the same purpose, considering that many of the reported transistors operate at moderate or large voltages (>10 V, which is preventing their use for practical applications, where biases of the order of a couple of volts would be required), to take into account the voltage ranges employed to characterize the devices, we decided to introduce the ratio of the I_{ON}/I_{OFF} values and the supply voltages V_{DD} in the x axis.

Following the Reviewer's suggestion, we have also compared our results, not only with devices fabricated on paper, but also with those obtained for other devices fabricated on different flexible substrates. However, we would also like to remark that, although they are both flexible, the surface chemistry and roughness of plastic and paper can be very different, so a direct comparison can be misleading.

As underlined in R#1_A1, we modified S5. *Comparison with the literature* section of the supplementary information including Figure S8 of the new manuscript and its respective table. Even if we appreciate the value of the works highlighted by the Reviewer, we chose not to include them in the comparison which is focused on flexible MoS₂-based FETs. Choi's results are not on flexible substrates, but on SiO₂/Si, as reported in Figure 2 and Table S1 of their manuscript. Even in this case, the I_{ON}/I_{OFF} ratio would have been smaller as compared to what reported and in particular equal to 10⁴. (Choi, M. *et al.* Flexible active-matrix organic light-emitting diode display enabled by MoS₂ thin-film transistor. *Sci. Adv.* **4**, 1–8 (2018)). Gong *et al.* presented WS₂ based FETs on Kapton characterized by mobility and I_{ON}/I_{OFF} values, of up to 11 cm² V⁻¹ s⁻¹ (with an average of 2 cm² V⁻¹ s⁻¹) and 10³, comparable to what is reported in our manuscript. However, these devices were fabricated entirely using micro-fabrication processes, such as thermal evaporation for the metal contacts and atomic layer deposition for the gate dielectric (Gong, Y., Carozo, V., Li, H., Terrones, M. & Jackson, T. N. High flex cycle testing of CVD monolayer WS₂ TFTs on thin flexible polyimide. *2D Mater.* **3**, 0–6 (2016)). As underlined by the

Reviewer, the adaptability of the inks employed in our work to another printing system has been recently demonstrated by two of the authors in (Lu, S. *et al.* Flexible, Print-in-Place 1D–2D Thin-Film Transistors Using Aerosol Jet Printing. *ACS Nano* **13**, 11263–11272 (2019), where hybrid CNT-2D material aerosol-jet printed transistors were reported mainly on Kapton and an example was also reported on paper. On Kapton, a maximum mobility of $10.7 \text{ cm}^2 \text{ V}^{-1} \text{ s}^{-1}$, an average mobility of $4.0 \text{ cm}^2 \text{ V}^{-1} \text{ s}^{-1}$ and $(I_{ON}/I_{OFF})/V_{DD}$ ratio of 1000 were obtained (in line with our results, but not on paper), while lower performance was registered on paper as shown in Figure S12 of the manuscript, where, for example, an $(I_{ON}/I_{OFF})/V_{DD}$ ratio smaller than 100 was obtained.

As can be seen from the comparison in Figure S8 of the revised version of the manuscript, our devices, ***despite being printed on paper, show performance comparable or even better than those of other devices fabricated on other flexible substrates***, whose performance are boosted by encapsulation layers, and further post-processing allowed by the larger thermal budget, not possible on paper. Better performance is shown by devices in the grey area, that anyway are produced through mechanical exfoliation of two-dimensional materials, which cannot be considered as a valid option from a large-scale fabrication point of view.

Changes made to the manuscript

- We have now added the work presented by Gong *et al.* (Gong, Y., Carozo, V., Li, H., Terrones, M. & Jackson, T. N. High flex cycle testing of CVD monolayer WS₂ TFTs on thin flexible polyimide. *2D Mater.* **3**, 0–6 (2016)) and by Choi *et al.* (Choi, M. *et al.* Flexible active-matrix organic light-emitting diode display enabled by MoS₂ thin-film transistor. *Sci. Adv.* **4**, 1–8 (2018)) to the references (Ref. 19 and Ref. 26, respectively).
- We modified Figure 2e and the relative comment in the main text (accordingly Table S1 of the supplementary information was updated):

Page 10: “Figure 2e shows μ_{FE} and I_{ON}/I_{OFF} for our devices compared with those previously reported in the literature: the closer the points to the top-right corner, the better the performance. We considered only transistors fully fabricated on paper or transferred on paper after fabrication. **For a fair comparison, all the I_{ON}/I_{OFF} values are re-calculated considering the International Technology Roadmap for Semiconductors (ITRS) definition³⁸, and then divided by the supply voltage V_{DD} . This normalization allows to take into account the operating voltage ranges of the considered FETs, which is a crucial problem for portable applications, where low power consumption is often required.** They have been divided into four groups according to the nature of the semiconductor used as channel: 2D materials^{24,28,42}, organic semiconductors^{43,44,45,46,47,48,49}, inorganic oxide semiconductors^{50,51,52,53,54}, carbon nanotubes (CNTs)^{55,56}. **Our devices show competitive electrical performance and are the only one, where both the contacts and the insulating layers are deposited by means of inkjet-printing** (for a detailed comparison see Table S1 in *Supplementary Information*). While maintaining high I_{ON}/I_{OFF} ratios, the mobility values

extracted from the MoS₂ FETs are, generally, larger than those obtained for organic semiconductors. Transistors that show comparable or better performance than ours, as reported in references ^{25,50,52,54}, were fabricated using micro-fabrication techniques for the deposition of insulator and contact layers, as well. It is worth mentioning that the mobility extracted in this work is comparable to the one found for CVD MoS₂ FETs entirely fabricated using conventional microelectronic techniques on plastic planarized paper substrates ²⁵. In most of the works, planarization layers were introduced to mitigate the surface roughness of the paper substrates and high-temperature processes were employed. Both of these aspects represent an increase in the complexity of the transistor manufacturing. As explained in *Methods*, except the deposition of the active layer, here, the fabrication process and the electrical characterization are carried out at ambient condition on commercially available paper substrates, that cannot withstand temperatures higher than 120°C. Moreover, no encapsulation layers are introduced. The potential of our approach stands in the coherent combination of two large-area fabrication processes in order to obtain good electrical performance. In *Supplementary Information* (see *Section S5.2*), a comparison with devices fabricated on flexible substrates (other than paper) is reported. As can be seen, our devices are comparable with the best-in-class presented devices.”

Figure 2e: Field-effect mobility and $(I_{ON}/I_{OFF})/V_{DD}$ for FETs characterized on paper substrates previously reported in the literature. V_{DD} is the supply voltage for each device. Blue stars, this work, inkjet-printed silver contacts; purple star, this work, inkjet-printed graphene contacts; black dots, 2D materials (25,28,42); red dots, organic semiconductors (43,44,45,46,47,48,49); yellow dots, inorganic oxides (50,51,52,53,54); green dots, CNTs (55,56).

- In section **S5. Comparison with the literature**. We have added a comparison with the literature, Figure S8 and Table S2 report a comparison between MoS₂-based FETs characterized on commonly-employed, flexible, plastic substrates, such as polyimide, PEN, or PET.

Page S10: “**S5.2 Comparison with MoS₂-based FETs on flexible substrates.**

Figure S8 shows μ_{FE} and $(I_{ON}/I_{OFF})/V_{DD}$ for our devices compared with those previously reported in the literature: the closer the points to the top-right corner, the better the performance. MoS₂-based transistors fully fabricated on flexible substrates or transferred on flexible substrates after fabrication are considered. The I_{ON}/I_{OFF} values are re-calculated considering the ITRS definition ^{S15}, and then divided by the V_{DD} . Table S2 shows the substrates, materials and deposition techniques, V_{GS} supply, μ_{FE} , and $(I_{ON}/I_{OFF})/V_{DD}$ for the manuscripts reported in Figure S8. The devices have been divided into 3 groups, according to the MoS₂ fabrication process.

The key challenge for the development of high-performing flexible electronics is the use of optimal semiconductors that show good mechanical flexibility, that can be processed using low-temperature approaches, and, most importantly, exploited in large-scale integrated circuits ^{S16}. For these reasons, devices in the grey area ^{S16,S17,S18,S19,S20,S21,S22}, which all present remarkable performance and are fabricated using mechanical exfoliated MoS₂, cannot be considered as a valid option.

Although devices in the green area do not show competitive performances, they have been included in this graph because they are all fabricated using quite challenging deposition techniques for the semiconductor ^{S23,S24,S25,S26,S27}. In 2017, for example, Kelly *et al.* ^{S23}, demonstrated a low-voltage FET characterized by a mobility of 0.15 cm² V⁻¹ s⁻¹ and I_{ON}/I_{OFF} value of 10, using spray coating for the deposition of the active layer.

All the transistors reported in the pink area are fabricated with CVD MoS₂ semiconducting layers ^{S13,S28,S29,S30,S31,S32,S33,S34,S35,S36,S37,S38}. As can be seen, our devices are well placed in terms of performance when compared to the others and they are the only ones that simultaneously present inkjet-printed insulating and contacts layers, are fabricated on paper, and can be operated using low-voltage.”

Figure S8 | Field-effect mobility and $(I_{ON}/I_{OFF})/V_{DD}$ for FETs characterized on flexible substrates previously reported in the literature. V_{DD} is the supply voltage for each device. Blue stars, this work, inkjet-printed silver contacts; purple star, this work, inkjet-printed graphene contacts; grey dots, mechanical exfoliated MoS₂ (S16, S17, S18, S19, S20, S21, S22); green dots, other deposition methods (S23, S24, S25, S26, S27); fuchsia dots, CVD MoS₂ (S13, S28, S29, S30, S31, S32, S33, S34, S35, S36, S37, S38). A detail comparison is reported in Table S2.

R#1_Q8: On page 9 the authors undertake bending tests to measure the flexibility of their devices, however there is no mention in the draft of the experimental set up making it difficult to reproduce. A description should be added to the methods section. Moreover, the thickness of the substrate used should be described so that the reader can calculate the strain that is applied to the device.

R#1_Q8: We thank the Reviewer for the useful suggestion, we added a paragraph in the supplementary information with the description of the bending test setup and the information that were missing.

Changes made to the manuscript

- Supplementary information (page S8):

“ S4.1 Bending test

To test the devices under tensile strain conditions, the paper substrate is wrapped around rigid jigs of different radii (R: 32, 20, 12, and 8 mm) and the electrical performance are characterized using the same setup (probe tips, ambient condition) reported in *Method*. The tensile strain S can be calculated using the following equation ^{S12,S13}:

$$S = \frac{t_{MoS_2} + t_{total}}{2R} \times 100 \quad (5)$$

Where t_{MoS_2} , t_{total} , and R are the thickness of the MoS₂ layer, the thickness of the device, and the bending radius, respectively. The paper employed in this work is characterized by a thickness of 275 μm ; thus, t_{total} can be considered equal to the substrate thickness. Tensile strains of 0.43%, 0.69%, 1.15%, and 1.72% are obtained from eq (5).”

R#1_Q9: There are also typo's in some the reference such as reference 5, 13, 18 and 51.

R#1_A9: We corrected the typos. All the references are now written in the correct way.

R#1_Q10: In my opinion, the manuscript is rational but flawed. The authors use an expensive process (CVD) with many fabrication steps with a cheap, fast and scalable technique (inkjet printing). The resulting devices are now expensive to manufacture, with lower figures of merit (mobility and on/off ratio) compared to CVD MoS₂ devices on flexible substrates and also have additional fabrication steps making the technology difficult to scale. The figures of merit presented for mobility and on/off ratio are unremarkable on a flexible substrate.

I recommend to reject this paper on the basis of novelty and unremarkable performance. Scientifically the paper is correct and would therefore recommend publication in a lower impact journal after revision.

R#1_A10: We thank the Reviewer for his/her comment, as it gives us the opportunity to provide more details regarding the novelty of our work, which was clearly not well delivered, based on the Reviewer comment. The rationale behind our channel array system is the following: whilst the choice of a cheap, additive, mask-less technique such as inkjet printing for the development of a paper based electronic system might be obvious, the introduction of the more expensive CVD-grown semiconductor layers could be seen as counter-productive. Inkjet-printing, however, presents critical issues that have, so far, limited the development of high-quality semiconducting layers, which are necessary in any transistor structure. So far, the best field-effect mobility and I_{ON}/I_{OFF} ratio reported for a TMD ink-based transistor are around $10 \text{ cm}^2 \text{ V}^{-1} \text{ s}^{-1}$ and $10 \text{ cm}^2/\text{Vs}$ and $(I_{ON}/I_{OFF})/V_{DD}$ around 100, respectively, but through solution-process methods and not inkjet (Lin, Z. et al. Solution-processable 2D semiconductors for high-performance large-area electronics. *Nature* **562**, 254–258 (2018)), which further need post-processing incompatible with paper substrate. Hence, until the issues associated with fully solution-process printed transistors are not solved, an alternative fabrication technique is strongly needed.

The combination of these methods, although well developed as individual techniques, is, however, far from trivial: the optimization of the ink formulation and rheological properties, and the control of the interaction between the ink and the substrate, do not necessarily result in a successful print. Similarly, applying the printing parameters, which were found to be the optimized values for small films, does not automatically result in a good film formation for crystalline materials. Moreover, printing a liquid dielectric on CVD material may affect the transport properties of the crystalline channel.

The CVD method is currently expensive, though our approach is cheaper than conventional CVD requiring only a quartz tube oven (which costs in the region of €20,000) and solid precursors (Dumcenco, D. et al. Large-Area Epitaxial Monolayer MoS₂. *ACS Nano* **9**, 4611–4620 (2015)), and may become the only solution if the issues associated to printed 2D transistors cannot be solved. Up to now, CVD is surely the most promising bottom up method for the large area synthesis of high-quality TMDs, especially considering the recent progress in the CVD growth of MoS₂ involving a low-cost, large-area roll-to-roll approach (Lim, Y. R. et al. Roll-to-Roll Production of Layer-Controlled Molybdenum Disulfide: A Platform for 2D Semiconductor-Based Industrial Applications. *Adv. Mater.*

30, 1705270 (2018)). It allows control of the layer number, thickness, domain size and morphology of the deposited layers.

The fabrication approach demonstrated in our manuscript allows the manufacturing of printed FET transistors on paper, working with low supply voltages (smaller than 2 V), and with electrical performance comparable to those of flexible FETs developed using micro-fabrication techniques (see details on device performance provided above). Furthermore, it allows the “on demand” fabrication of more complex circuits through the ASIC design philosophy, providing full design flexibility of the circuit.

Hence, we believe that our work does not deserve rejection only based on costs, which may change in future and may be justified by the final application. We would like to invite the Reviewer to also look at the novelty and impact produced by our results, both from the device physics and engineering point of views.

Based on the above statements, we do strongly believe that our approach is far from being incremental, as also remarked by Reviewer #2, who define our approach “smart” and compatible with customization (*“I also recommend to highlight the quite smart approach of customizing the transistors by inkjet printing which allows a mass production of not customized/potentially not patterned semiconducting channel layers followed by customizing the transistors using inkjet printing.”*) and Reviewer #3.

Customization is indeed another aspect that we have highlighted in the revised manuscript, following advice from Reviewer 2 (see related answer).

Reviewer #2 (Remarks to the Author):

R#2: The presented work is very interesting and the performance of the devices is really promising. The device fabrication and characterization has been done properly and with a lot of expertise. The manufacturing approach is novel and smart and is of interest to others in the community and the wider research field.

We thank the Reviewer for the appreciation of our work and for pointing out the degree of novelty of our fabrication approach.

R#2 _Q1: I would recommend to highlight the manufacturing approach and in detail the combination of both manufacturing steps/methods (CVD + inkjet) with their respective technical advantages and disadvantages - also in terms of the potential future industrial use of these technologies for the manufacturing of such devices and the need of patterning the semiconductor.

I also recommend to highlight the quite smart approach of customizing the transistors by inkjet printing which allows a mass production of not customized/potentially not patterned semiconducting channel layers followed by customizing the transistors using inkjet printing.

R#2_A1: We thank the Reviewer for the appreciation shown towards our idea of customizing and designing field effect transistors through inkjet-printing, while exploiting pre-patterned CVD MoS₂ channels. In order to comply with Reviewer's suggestions, we have now modified the manuscript to strengthen the importance of our fabrication approach, also highlighting the advantages and disadvantages of the different fabrication techniques.

Changes made to the manuscript

- The novelty of the proposed approach has been highlighted in the revised version of the introduction:

Page 3: **“We combine chemical vapour deposition (CVD), for the growth of high-quality MoS₂ channels, with inkjet-printing, which gives the possibility to design and fabricate customizable devices and circuits exploiting 2DMs-based inks, whose capability to be printed on top of CVD grown materials has been successfully demonstrated in ¹⁶. In this way, an application specific integrated circuit (ASIC) design approach, known as “channel array”, is proposed. It is based on the transfer of strips of CVD-grown MoS₂, the transistor channels, onto a paper substrate where the rest of the devices and circuits, source and drain contacts (defining the channel length and width), gate dielectric, gate contacts, and connections, are fully customized exploiting the inkjet-printing technique, giving a further degree of freedom to the designer.** This method allows to keep the flexibility and versatility of an all-inkjet technology, with the difference that here a high-quality channel is already placed on the substrate, by taking advantage of the CVD-grown TMDC. **Moreover, both being bottom up, large-area**

fabrication processes, their combination could open a possible exploitation at the industrial level.”

~~We propose a new fabrication technique that combine the use of high mobility MoS₂ channels grown by CVD and inkjet printing technique for the fabrication of transistors and circuits, exploiting 2DMs based inks, whose capability to be printed on top of CVD grown materials has been successfully demonstrated in¹⁶. This approach is called “channel array”, as it is based on the deposition of several channels on a customized substrate, onto which the rest of the device and circuit is fabricated. The “channel array” technique consists of two steps: first, the channel array is fabricated by transferring on the paper substrate stripes of CVD grown MoS₂, which will be used as transistors channels. In the second step, the FETs and the rest of the circuit are fabricated by customizing the channel array by inkjet printing the source and drain contacts (i.e., channel length and width), gate dielectric, gate contacts, and all connections. This approach allows to keep the flexibility and versatility of an all inkjet technology, with the difference that here a high quality channel is already placed on the substrate, by taking advantage of the CVD grown TMDC.~~

- We have modified the beginning of Fabrication of MoS₂ FETs paragraph of the manuscript, in order to underline the advantages and disadvantages of the considered technique:

Page 5: **“The rationale of our approach is the combination of two bottom up fabrication techniques to have high-quality semiconducting substrates easily customizable to obtain device and circuits with a versatile printing technique. The advantage of inkjet-printing is the fast prototyping, which allows for on-the-fly corrections as well as easy pattern changes, simplifying the manufacturing process. Moreover, being an additive and mask-less method, it also cuts down materials and energy consumption, reducing the number of processing steps, time, space, and waste production during the fabrication. On the other hand, inkjet-printing presents critical aspects, such as the need to use inks with specific rheological properties, and, more importantly, the current lack of semiconducting 2DM-based inks for high performance FETs. Even if expensive, lacking in compatibility with arbitrary substrates, suffering from atomic vacancies and batch-to-batch variations, CVD is so far the most-promising bottom-up approach in order to obtain high-quality semiconducting layer and may become the method of choice, considering the recent progress in the CVD growth of MoS₂ involving a low-cost and very large area roll-to-roll approach²⁹. Thus, it was introduced as a supplementary fabrication technique.”**

R#2 _Q2: Related to this statement: "However, paper is a challenging substrate for electronics due to its high roughness and limiting processing temperature and the lack of a winning fabrication technique is preventing its exploitation at the industrial level." I think it is important to add another 1-3 sentences with literature sources (e.g. <https://onlinelibrary.wiley.com/doi/abs/10.1002/adma.201801588>) to this statement in order to indicate some additional challenges of the concept of paper electronics. Next to

roughness and temperature stability, I think it is important to give some indication about further potential limitations/challenges, e.g. about dimensional stability and durability (e.g. as a function of humidity, temperature etc.), functionalization/priming/coatings of paper surfaces, and about reproducibility of paper and paper surface properties (since paper is a natural material).

In addition, I recommend to add more technical details (surface roughness, material composition, ...) about the used paper from PEL which seems to be a very special paper with a special microporous top coat?

R#2_A2: We thank the Reviewer for the valuable comments. We agree that a more detailed discussion on paper electronic can give the reader a more complete perspective on this challenging new technology.

Changes made to the manuscript

- In order to highlight the challenges related to the use of a paper substrate, the main text has been modified as follows:

Page 2: “Derived from abundant and renewable raw materials, paper-based consumer electronics is expected to alleviate landfill and environmental problems and to reduce the impact associated with recycling operations, whilst offering cost-effectiveness, easy scalability, and large flexibility ³. **Despite the fact that several devices and applications have been reported in the literature ⁴, paper is still a challenging substrate for electronics, rarely employed without the addition of coating/laminating layers ^{5,6}. Its porous structure (which in turn leads to high roughness), limited stability and durability (mainly due poor thermal and humidity resistance), and high hygroscopicity (which can influence the electrical characterization), combined with the lack of winning reliable fabrication techniques, is preventing its exploitation at the industrial level ^{7,8}.”**

~~However, paper is a challenging substrate for electronics due to its high roughness and limiting processing temperature and the lack of a winning fabrication technique is preventing its exploitation at the industrial level.~~

- We added technical details about the employed paper substrate in the Supplementary Information, **S6. PEL P60 details:**

Page S14: **PEL P60 (purchased from Printed Electronics Limited) is a cellulose-based paper (of a thickness of around 250 µm) coated with a microporous ceramic slurry (of a thickness of around 25 µm). Figure S9 shows the profile of the paper surface.**

Figure S9 | Variations of the surface of the paper substrate employed in this work.

R#2_Q3: Please provide a short comparison performance wise to printed transistors on polymer films - either based on your studies (maybe you have also tried to use polymer films instead of paper?) or to literature studies.

R#2_A3: Being paper the target substrate of this work, we did not try any polymer substrate for the fabrication of our devices. We also remark that, although being both flexible, the surface chemistry and roughness of plastic and paper can be very different, so a direct comparison can be misleading.

Regarding the point raised by the Reviewer, among 2D material-based FETs, Kim *et al.* (Kim, T. Y. *et al.* Transparent Large-Area MoS₂ Phototransistors with Inkjet-Printed Components on Flexible Platforms. *ACS Nano* **11**, 10273–10280 (2017)) presented a hybrid organic-2D material transistors on alumina coated PET substrate with inkjet-printed insulator/contact layers. In that case, a mobility of $0.37 \text{ cm}^2 \text{ V}^{-1} \text{ s}^{-1}$ and I_{ON}/I_{OFF} ratio of around 10 were obtained. However, this work is not directly comparable with our research which is focused on different materials tested at lower voltage ranges on a less mature and more challenging substrate. In 2017, Kelly *et al.* (Kelly, A. G. *et al.* All-printed thin-film transistors from networks of liquid-exfoliated nanosheets. *Science* **356**, 69–73 (2017)), demonstrated a low-voltage MoS₂ FET characterized by a mobility of $0.15 \text{ cm}^2 \text{ V}^{-1} \text{ s}^{-1}$ and I_{ON}/I_{OFF} value of 10, using spray coating for the deposition of the active layer. They showed similar results for MoSe₂, WS₂, and WSe₂. Fully printed graphene FETs with a maximum field-effect mobility of $91 \text{ cm}^2 \text{ V}^{-1} \text{ s}^{-1}$ were presented by Carey *et al.* (Carey, T. *et al.* Fully inkjet-printed two-dimensional material field-effect heterojunctions for wearable and textile electronics. *Nat. Commun.* **8**, 1202 (2017)). The absence of a band gap in the material, however, gives low I_{ON}/I_{OFF} values which negatively affect the possibility of using these devices for logic applications.

Thanks to the possibility of being easily processed from solution, combined with flexibility, and ease of functionalization, organic materials, have been, so far, the main object of printed electronics research area (Fukuda, K. & Someya, T. Recent Progress in the Development of Printed Thin-Film Transistors and Circuits with High-Resolution Printing Technology. *Adv. Mater.* **29**, (2017)). In the last decade, in

the effort of improving transistors' performance (often characterized by low mobilities values and non-reliable I_{ON}/I_{OFF} values), various solution processing fabrication techniques to enhance the organic semiconductor crystallinity have been reported. However, complicated manufacturing approaches and performance inhomogeneities still hamper their effective applicability and commercialization. In 2014, for example, Fukuda *et al.* (Fukuda, K. *et al.* Fully-printed high-performance organic thin-film transistors and circuitry on one-micron-thick polymer films. *Nat. Commun.* **5**, 4147 (2014)) reported fully printed transistors and circuits based on a commercial p-type organic semiconductor fabricated on 1-mm-thick parylene-C films with an average field-effect mobility of around $0.34 \text{ cm}^2\text{V}^{-1}\text{s}^{-1}$ (best result $1 \text{ cm}^2\text{V}^{-1}\text{s}^{-1}$) and operating speeds (about 1 ms) at low operating voltages. However, the fabrication process consisted in more than 20 steps and involved the use of fluoropolymer banks to delimitate the spreading of the semiconductor over the underlying layers of the devices.

R#2_Q4: Please introduce the dielectric Hexagonal Boron Nitride so that the abbreviation hBN becomes more clear.

R#2_A4: We thank the Reviewer for pointing this out. We have now introduced the full name in the manuscript (page 7).

R#2_Q5: When using silver or graphene inks for the S-D and G electrodes: What do you think about the influence of work function? What was the post-processing (e.g. annealing)?

R#2_A5: No post-processing treatment or annealing step was performed during any step of the fabrication of the transistors reported in the manuscript, exploiting either silver or graphene electrodes. It is important to point out that we usually waited at least 2 hours before printing the next component of the device (e.g., after having printed the insulator, we generally left the substrate on the printer platen for two hours at ambient condition, before printing the silver gate contact).

Regarding the work functions (WFs) and according to the literature, the WF of inkjet-printed silver electrodes ranges from around 3.7 eV to around 5 eV, as a function of the ink formulation, the layer formation characteristics, and the post-deposition processing (Tobjörk, D., Kaihovirta, N. J., Mäkelä, T., Pettersson, F. S. & Österbacka, R. All-printed low-voltage organic transistors. *Org. Electron. physics, Mater. Appl.* **9**, 931–935 (2008); Mitra, D. *et al.* Work Function and Conductivity of Inkjet-Printed Silver Layers: Effect of Inks and Post-treatments. *J. Electron. Mater.* **47**, 2135–2142 (2018)). The WF of undoped, defect-free graphene is strictly dependent on the nanosheet thickness, varying from 4.3 eV (below vacuum) for monolayers to 4.6 eV for nanosheets consisting of more than 10 monolayers (Hibino, H. *et al.* Dependence of electronic properties of epitaxial few-layer graphene on the number of layers investigated by photoelectron emission microscopy. *Phys. Rev. B* **79**, 125437 (2009)). However, as for silver, the WF of inkjet-printed graphene depends on several factors, including the ink fabrication technique and the employed solvents and binders. It follows that an estimation of the WFs of the inkjet-printed electrodes employed in this work is not trivial. In addition, Anderson's rule

does not apply, since, as we have indeed experienced, the contact/semiconductor junction is dominated by the presence of different type of defects, which causes pinning of the Fermi levels, and a simple picture of the energy barrier related to the difference between the WF of the metal and the electron affinity of the semiconductors does not hold (Gong, C. *et al.* Band alignment of two-dimensional transition metal dichalcogenides: Application in tunnel field effect transistors. *Appl. Phys. Lett.* **103**, 053513 (2013); McDonnell, S., Addou, R., Buie, C., Wallace, R. M. & Hinkle, C. L. Defect-Dominated Doping and Contact Resistance in MoS₂. *ACS Nano* **8**, 2880–2888 (2014); Cusati, T. *et al.* Electrical properties of graphene-metal contacts. *Sci. Rep.* **7**, 5109 (2017); Soththewes, K. *et al.* Universal Fermi-Level Pinning in Transition-Metal Dichalcogenides. *J. Phys. Chem. C* **123**, 5411–5420 (2019)).

In order to satisfy Reviewer’s request, we have investigated the effect of thermal annealing on the MoS₂ FETs with inkjet-printed silver contacts. In particular, Figure R#2_A5a shows the transfer characteristic of an individual MoS₂ FET on paper exploiting inkjet-printed Ag contacts, carried out prior and after thermal annealing in vacuum (10⁻² mbar) at 100 °C, for 30 minutes (this temperature has been chosen considering the limitations introduced by the paper substrate used in this work). After thermal annealing, the FET transfer characteristic becomes steeper, pointing out an increased charged carrier mobility. We believe that the reduced thermal budget provided during annealing is not sufficient for reorganization of the crystalline lattice of the MoS₂ channel, which was grown at a much higher substrate temperature of 750 °C. The increased mobility is attributed to either desorption of H₂O and O₂ molecules from the MoS₂ surface, which act as traps for electrons in the channel (Davis, S. M. & Carver, J. C. Oxygen chemisorption at defect sites in MoS₂ and ReS₂ basal plane surfaces. *Appl. Surf. Sci.* **20**, 193–198 (1984)), or to improvement of the Ag/MoS₂ contact. Due to the reversibility of the observed change (after about 3 days the FET transfer characteristic goes back to its original behavior) and to the observed effect of thermal annealing on the leakage current (Figure R#2_A5b), we believe that the first effect is the dominating one.

Figure R#2_A5: I_{DS} vs V_{GS} (a) and I_{GS} vs V_{GS} (b) of a MoS₂ FET with inkjet-printed silver contact before (black dots) and after (blue dots) an annealing treatment at 100°C, 10⁻²mbar, for 30 minutes, characterization of the same device after 3 days (red dots).

Changes made to the manuscript

The absence of annealing steps in the fabrication procedure has been underlined in *Methods*:

- Page 19: “**It is worth underlining that no annealing or post-treatment process is performed after any printing step.**”

R#2_Q6: *TFT vs FET - I recommend to go for one term only in order to have a more focused terminology.*

R#2_A6: We thank the Reviewer for this comment. We have now introduced FET as the only acronym in the manuscript.

R#2_Q7: *Could you please confirm that 1 pL and not 10 pL (nominal volumes) printheads were used for the deposition of the inks?*

R#2_A7: We confirm that we used 1pL cartridges to print silver contacts, whilst for the printing process of both graphene contacts and hBN insulating layer 10 pL cartridges have been used.

Changes made to the manuscript

- We added this aspect to the *Methods* in the main manuscript (page 19).

R#2_Q8: *Please report further details about the printed and CVD processes layer properties, especially (i) thickness/roughness of the dielectric layer after deposition of the layer in so many passes, (ii) W and L of the transistors for all the results/graphs, parameters of the CVD layer.*

R#2_A8: We thank the Reviewer for her/his useful comment. Regarding the morphological characterization of both MoS₂ and the dielectric on paper, we cannot provide precise information on the hBN thickness due to the large roughness of the paper substrate (please refer to Figure S10, previously cited in Answer R#2_A2). However, to give an idea of the printed dielectric layer thickness, we have measured the thickness of 80 hBN print passes on a glass substrate and we have obtained a thickness value of around 5 μm, which is in line with what previously reported by some of the present authors (Worsley, R. *et al.* All-2D Material Inkjet-Printed Capacitors: Toward Fully Printed Integrated Circuits. *ACS Nano* **13**, 54–60 (2019).). As a word of caution, we have to say that we are expecting a slightly different value on paper, because of the different surface tension, wettability, and porosity of the substrate.

The morphological characterization of the MoS₂ layer on a sapphire substrate, as well as a table showing the widths and lengths of the transistors have been included in the supplementary information.

Changes made to the manuscript

- *Methods*, page 20: “Several transistors have been fabricated (with a yield of around 80%) and characterized with a nominal width of ~ 500 μm and length varying between **40 μm and 60 μm (further details can be found in Table S3 of Section S7 in the Supplementary Information).**”
- A table with the geometrical dimensions of the 26 MoS₂ FETs is now reported in Section S7 of the Supplementary Information (page S15).

- Supplementary Information, page S4:

S2. Atomic force microscopy of MoS₂ film on sapphire.

Figure S4 shows an atomic force microscopy micrograph of the CVD-grown MoS₂ film, obtained from solid precursors, on sapphire before the transfer, characterized by a root mean square roughness of around 50 pm.

Figure S4 | Atomic force microscopy micrograph of the CVD MoS₂ film. The film is characterized on sapphire before the transfer. The images on the sides are zooms of 5 μm x 5 μm of surface area.

R#2_Q9: Could you please give more information about the manufacturing yield, some statistics (e.g. do you see any dependencies on W or L , what is the reason for 80% yield, which process is the most challenging process, ...), causes of the transistor performance variability reported in the appendix and impact on the performance of circuits (e.g. taking in account literature such as <https://www.sciencedirect.com/science/article/abs/pii/S1566119914000287>) and your opinion about further industrialization of the processes (e.g. taking in account literature studies of transistors completely manufactured by inkjet printing such as <https://www.nature.com/articles/srep33490>).

R#2_Q9: In relation to the discussion reported by Sowade *et al.* (Sowade, E. *et al.* All-inkjet-printed thin-film transistors: manufacturing process reliability by root cause analysis. *Sci. Rep.* **6**, 33490 (2016)), we would like to underline that, concerning the printing part of our fabrication process, the step that most negatively affected the manufacturing yield is represented by the deposition of the insulator layer. We registered negligible failures (less than 0.5%) related to short circuits between S/D/G contacts or open circuits along the single electrodes.

As widely reported in the literature (Cui, Z. *et al.* Printed Electronics: Materials, Technologies and Applications, First Edition, John Wiley & Sons, Singapore, 2016), the use of an inkjet-printed insulating

layer can be quite challenging, and it often represents the main issue affecting the fabrication process yield. The insulating film should present moderate surface roughness, high electrical field strength, ideally a high value of dielectric constant and withstand the impact of solvents in the subsequent process steps. The ink employed in this work has shown good qualities, it has a relative permittivity of around 6 and a breakdown field of around 2 MVcm^{-1} , as recently demonstrated (Worsley, R. et al. All-2D Material Inkjet-Printed Capacitors: Toward Fully Printed Integrated Circuits. *ACS Nano* **13**, 54–60 (2019)), where step-by-step annealing procedure at 150°C for 2 hours was employed. However, in our case, the incompatibility of this temperature with the paper substrates has forbidden the introduction of this procedure (confront R#2_A5). Indeed, the yield is mainly limited by large leakage currents (I_{Leak}) in some devices. As a criterion, we have excluded devices with $I_{Leak} > 5 \text{ nA}$. A possible explanation for the high I_{Leak} observed in some devices, is likely related to non-uniformities in the insulating layers, which may lead to the presence of pinholes in the printed film (Kelly, A. G., Finn, D., Harvey, A., Hallam, T. & Coleman, J. N. All-printed capacitors from graphene-BN-graphene nanosheet heterostructures. *Appl. Phys. Lett.* **109**, 023107 (2016)), in our case probably induced by the use of a paper substrate. To obtain robust insulating layers, the ink was deposited using 80 printing passes, with a drop spacing of $20 \mu\text{m}$, keeping the cartridge sweep at a frequency of 1 KHz, as done in Worsley, R. et al. All-2D Material Inkjet-Printed Capacitors: Toward Fully Printed Integrated Circuits. *ACS Nano* **13**, 54–60 (2019), where a similar yield was found.

Similarly, we have noticed that the yield depends on the device area ($A = WL$). The larger A , the higher the probability of having pinholes. The yield reported in the paper is based on devices with a channel width of $W \sim 500 \mu\text{m}$ and a length L ranging from $40 \mu\text{m}$ to $60 \mu\text{m}$. Keeping L constant, we reduced W in order to fabricate devices with smaller A . In particular, we fabricated 10 devices; 5 with an area reduced by a factor of 5; and 5 with an area reduced by a factor of 10 with respect to that investigated in the paper. A yield of 100% was achieved for these devices. The dependence of the yield on the area is in agreement with previous results reported for capacitors in Worsley et al. (Worsley, R. et al. All-2D Material Inkjet-Printed Capacitors: Toward Fully Printed Integrated Circuits. *ACS Nano* **13**, 54–60 (2019))

Another factor that has negatively affected the device yield is related to the quality of the semiconductor. First, the transfer of the CVD-grown semiconductor from the native substrate onto the paper could partially degrade the quality of the transferred semiconductor. Secondly, while variations in the gate capacity are due to the insulator, the threshold voltages and the mobility values are, in general, a function of the dielectric, the semiconductor, and their interface (Feng, W., Zheng, W., Cao, W. & Hu, P. Back Gated Multilayer InSe Transistors with Enhanced Carrier Mobilities via the Suppression of Carrier Scattering from a Dielectric Interface. *Adv. Mater.* **26**, 6587–6593 (2014)). The relatively high I_{OFF} values registered for some of the devices are probably related to the presence of traps at the semiconductor/insulator interface and of leakage current variations in the dielectric layer (Li, T., Wan,

B., Du, G., Zhang, B. & Zeng, Z. Electrical performance of multilayer MoS₂ transistors on high- κ Al₂O₃ coated Si substrates. *AIP Adv.* **5**, (2015). Figure S7a shows that the *ON* current varies from 1 μ A to nearly 50 μ A. This is probably related to inhomogeneities in the CVD MoS₂ films and to the presence of grain boundaries nearly randomly distributed across the MoS₂ film (Kim, T. Y. *et al.* Transparent Large-Area MoS₂ Phototransistors with Inkjet-Printed Components on Flexible Platforms. *ACS Nano* **11**, 10273–10280 (2017). Charge carriers interacting with shallow traps, possibly at the MoS₂/dielectric interface can be the cause, of the anticlockwise hysteresis shown in Figure S7a and, consequently, of the variation in the threshold voltage (the average value $V_{THforward}$ is $(0.38 \pm 0.07)V$, the average value $V_{THbackward}$ is 0.14 ± 0.06) (Shah, P. B. *et al.* Analysis of temperature dependent hysteresis in MoS₂ field effect transistors for high frequency applications. *Solid. State. Electron.* **91**, 87–90 (2014). Variations in the electrical parameters of MoS₂ transistor have also been observed for devices where the insulator is fabricated with highly controllable techniques (Illarionov, Y. Y. *et al.* Ultrathin calcium fluoride insulators for two-dimensional field-effect transistors. *Nat. Electron.* **2**, 230–235 (2019).), as well as the presence of hysteresis (Late, D. J., Liu, B., Matte, H. S. S. R., Dravid, V. P. & Rao, C. N. R. Hysteresis in single-layer MoS₂ field effect transistors. *ACS Nano* **6**, 5635–5641 (2012); Illarionov, Y. Y. *et al.* Improved Hysteresis and Reliability of MoS₂ Transistors with High-Quality CVD Growth and Al₂O₃ Encapsulation. *IEEE Electron Device Lett.* **38**, 1763–1766 (2017); Di Bartolomeo, A. *et al.* Hysteresis in the transfer characteristics of MoS₂ transistors. *2D Mater.* **5**, (2018). In addition, bias stress effects, which can induce a shift in the threshold voltage and a variability of the device parameters, have been previously observed in MoS₂ (Cho, K. *et al.* Electric stress-induced threshold voltage instability of multilayer MoS₂ field effect transistors. *ACS Nano* **7**, 7751–7758 (2013); Illarionov, Y. Y. *et al.* Energetic mapping of oxide traps in MoS₂ field-effect transistors. *2D Mater.* **4**, (2017).

The impact of the variability of transistor parameters on circuits is certainly an extremely important factor. However, the current reproducibility of our devices already enables us to design simple circuits, in particular digital circuits more robust to the dispersion of transistor parameters. We believe that, at this point, it is useful both to further optimize the fabrication process and to collect statistical information of the transistor parameters using a larger number of devices. This data will then be employed to develop Monte Carlo simulations, which can provide useful information from the circuit design point of view, as reported, for example, in the manuscript underlined by the Reviewer (Myny, K., Van Lieshout, P., Genoe, J., Dehaene, W. & Heremans, P. Accounting for variability in the design of circuits with organic thin-film transistors. *Org. Electron.* **15**, 937–942 (2014))

As largely underlined in this reply and in the manuscript, after the transfer of the MoS₂ stripes onto the paper substrates, is possible to print all the drain/source contacts of the required devices, on one or more stripes, at the same time, i.e. with a single print operation. This can be repeated for the following steps, i.e. the printing of the dielectric layers and the gate contacts. Printing all devices at the same time can

also lead to more uniformity of the printed layers, and therefore to a smaller variability of the device parameters. In this way an array of transistors can be obtained and the interconnections between the elements can be defined with large flexibility. Moreover, exploiting the possibility of depositing CVD-grown MoS₂ involving a low-cost and large area roll-to-roll approach as recently demonstrated by Lim *et al.* (Lim, Y. R. *et al.* Roll-to-Roll Production of Layer-Controlled Molybdenum Disulfide: A Platform for 2D Semiconductor-Based Industrial Applications. *Adv. Mater.* **30**, 1705270 (2018)), the fabrication process proposed in this manuscript could open up to a potential new large area production technology.

Changes made to the manuscript

- Device statistic has been largely improved with 26 devices fabricated and characterized.
- *Section S4. Detailed electrical characterization* of the supplementary information has now been updated.

Page S7: **In Figure S7 the electrical characterization of 26 MoS₂ FETs is reported. Figure S7a and Figure S7b show the transfer characteristics and the gate current vs gate voltage curves for each device, respectively. A yield of 80% was obtained. As a criterion, we have excluded devices with $I_{Leak} > 5$ nA and I_{ON}/I_{OFF} smaller than 10^3 . A possible explanation for the high I_{Leak} observed in some devices, is likely related to non-uniformities in the insulating layers, which can lead to the presence of pinholes in the printed layer ^{S11}.**

The distribution of the threshold voltages and the field-effect mobility values in the forward sweep are reported Figure S7c and Figure S7d, respectively. The distribution of the threshold voltages and the field-effect mobility values in the backward sweep are reported Figure S7e and Figure S7f, respectively. The variability of the threshold voltages and the mobility values are, in general, a function of the dielectric, the semiconductor, and their interface ^{S12}. The on currents vary from 1 μ A to nearly 50 μ A. This is probably related to inhomogeneities in the CVD MoS₂ films and the presence of MoS₂ grain boundaries ^{S13}. Charge carriers interacting with shallow traps, possibly at the MoS₂/dielectric interface can be the cause, of the anticlockwise hysteresis shown in Figure S7a and, consequently, of the variation in the threshold voltage (the average value $V_{THforward}$ is (0.38 ± 0.07) V, the average value $V_{THbackward}$ is (0.14 ± 0.06) V.

Figure S1 | Electrical characterization of 26 MoS₂ FETs. a, Transfer characteristic curves. b, Gate current vs gate voltage curve. c, Distribution of threshold voltage values (forward). d, Distribution of field-effect mobility values (forward). e, Distribution of threshold voltage values (backward). f, Distribution of field-effect mobility values (backward).

R#2_Q10: Please provide some details about the "printability/jettability" of the inks. Is the printing performance stable? Could you apply the inks to other, more productive printheads?

R#2_Q11: We did not observe any problem regarding the printability or the jettability of the inks we used, and the printing performance was stable throughout the process. We would like to underline that no pre-treatment of the substrate was carried out, and the whole printing process was performed under ambient conditions. As it is reported in Methods, we employed a commercial silver ink provided by Sigma Aldrich to print the silver contacts (details can be found in Methods). The in-house graphene and

hBN inks were prepared following procedures previously developed by some of the present authors (McManus, D. et al. Water-based and biocompatible 2D crystal inks for all-inkjetprinted heterostructures. *Nat. Nanotechnol.* **12**, 343–350 (2017). The formulations of the inks have been optimized for the piezoelectric inkjet printer used in this work. A full characterization of the materials, the optimization of the formulations and the printing processes can be found in McManus, D. et al. Water-based and biocompatible 2D crystal inks for all-inkjetprinted heterostructures. *Nat. Nanotechnol.* **12**, 343–350 (2017) and Worsley, R. et al. All-2D Material Inkjet-Printed Capacitors: Toward Fully Printed Integrated Circuits. *ACS Nano* **13**, 54–60 (2019). In the future, it would be very interesting to test the inks with industrial inkjet printers. The adaptability of the inks to different printing systems has been recently reported by our colleagues in Lu, S. et al. Flexible, Print-in-Place 1D–2D Thin-Film Transistors Using Aerosol Jet Printing. *ACS Nano* **13**, 11263–11272 (2019). The rheological properties of these formulations have been successfully modified, through the addition of propylene glycol, as a secondary solvent, and hydroxypropyl methylcellulose, as a binder, to make them compatible with an aerosol jet printing system.

R#2_Q11: Did you see any problems with the different interfaces during the liquid processing (printing), e.g. dewetting effects?

R#2_Q11: As reported above (R#2_A5), even if during the fabrication process, we did not perform any annealing step on any layer of the transistors, we usually waited at least 2 hours before printing the next component of the device. We did not observe any de-wetting effect during the printing process. We tried to print the same structure using an organic insulator (poly(4-vinylphenol)) without performing any annealing step: in this case the de-wetting effect is very evident.

Reviewer #3 (Remarks to the Author):

R#3: The manuscript is interesting and reports advances of 2d devices on paper platform.

We thank the Reviewer for her/his comment on our paper, underlining the advances provided by our work.

R#3_Q1: Paper is a low-cost substrate ...however, the technique of integration is quite expensive. Is this really an attractive option compared to fully printed or solution based methods of realizing paper electronics?

R#3_A1: We thank the Reviewer for the valuable comment, which allows to provide further clarification on why our approach is new and of high importance for the community.

As already pointed out in one of our previous points, the simple fabrication we propose gives the opportunity to develop “on demand” complex circuits through the ASIC design philosophy, using high-quality semiconducting substrates, still challenging to obtain from solution-processing methods, that can be easily customized to result in performing thin film devices with the versatile technique of inkjet-printing. The rationale behind our channel array system is the following: whilst the choice of a cheap, additive, mask-less technique such as inkjet printing for the development of a paper based electronic system might be obvious, the introduction of the more expensive CVD-grown semiconductor layers could be seen as counter-productive. Inkjet-printing, however, presents critical issues that have, so far, limited the development of high-quality semiconducting layers, which are necessary in any transistor structure. So far, the best field-effect mobility and I_{ON}/I_{OFF} ratio reported for a TMD ink-based transistor are around $10 \text{ cm}^2 \text{ V}^{-1} \text{ s}^{-1}$ and $10 \text{ cm}^2/\text{Vs}$ and $(I_{ON}/I_{OFF})/V_{DD}$ around 100, respectively, but through solution-process methods and not inkjet (Lin, Z. *et al.* Solution-processable 2D semiconductors for high-performance large-area electronics. *Nature* **562**, 254–258 (2018)), which further need post-processing incompatible with paper substrate. Hence, until the issues associated with fully solution-process printed transistors are not solved, an alternative fabrication technique is strongly needed.

The combination of these methods, although well developed as individual techniques, is, however, far from trivial: the optimization of the ink formulation and rheological properties, and the control of the interaction between the ink and the substrate, do not necessarily result in a successful print. Similarly, applying the printing parameters, which were found to be the optimized values for small films, does not automatically result in a good film formation for crystalline materials. Moreover, printing a liquid dielectric on CVD material may affect the transport properties of the crystalline channel.

The CVD method is currently expensive, though our approach is cheaper than conventional CVD requiring only a quartz tube oven (which costs in the region of €20,000) and solid precursors (Dumcenco, D. *et al.* Large-Area Epitaxial Monolayer MoS₂. *ACS Nano* **9**, 4611–4620 (2015)), and may become the only solution if the issues associated to printed 2D transistors cannot be solved. Up to now, CVD is surely the most promising bottom up method for the large area synthesis of high-quality

TMDs, especially considering the recent progress in the CVD growth of MoS₂ involving a low-cost, large-area roll-to-roll approach (Lim, Y. R. *et al.* Roll-to-Roll Production of Layer-Controlled Molybdenum Disulfide: A Platform for 2D Semiconductor-Based Industrial Applications. *Adv. Mater.* **30**, 1705270 (2018)). It allows control of the layer number, thickness, domain size and morphology of the deposited layers.

The fabrication approach demonstrated in our manuscript allows the manufacturing of printed FET transistors on paper, working with low supply voltages (smaller than 2 V), and with electrical performance comparable to those of flexible FETs developed using micro-fabrication techniques (see details on device performance provided above). Furthermore, it allows the “on demand” fabrication of more complex circuits through the ASIC design philosophy, providing full design flexibility of the circuit.

Changes made to the manuscript

- In order to stress the degree of novelty and the obtained results, we have extensively changed the text (Please refer to R#2_A1)

R#3_Q2: the circuits are non-cmos ...I seriously doubt that RTL logic will be suitable in this modern era. Can the authors demonstrate cmos logic by using additional TMDs for p-type?

R#3_A2: We thank the Reviewer for this comment. We would like to underline that the main subject and novelty of our work concerns the fabrication of printed field-effect transistors on paper, working with low supply voltages and showing good electrical performance. The logic gates reported in the manuscript and fabricated exploiting resistor-transistor logic (RTL) technology should be considered as demonstrators of the actual applicability of our transistors for digital applications, and not as the architecture to choose, due to the well-known issues of static power dissipation and delays associated with the RTL solution. Clearly, CMOS would represent a better option, but we want to stress that obtaining CMOS technology with two-dimensional materials is already an issue when using traditional microfabrication techniques on rigid substrates. For example, Pu *et al.*, reported an inverter on a polyimide substrate, using quasi-CMOS technology, i.e. exploiting a single ambipolar TMD (Pu, J. *et al.* Highly flexible MoS₂ thin-film transistors with ion gel dielectrics. *Nano Lett.* **12**, 4013–4017 (2012)). Recently, Lee *et al.* presented a fully-CMOS using MoS₂ and WSe₂ on a PET substrate. However, the whole circuit was previously fabricated onto a rigid substrate using traditional microelectronic techniques and then transferred onto the flexible one (Lee, H. *et al.* Transfer of transition-metal dichalcogenide circuits onto arbitrary substrates for flexible device applications. *Nanoscale* **11**, 22118–22124 (2019)).

The suggestion from the Reviewer is definitively very attractive and would deserve a publication on its own. However, its achievement is beyond the scope of this work. In this perspective, our results are starting points towards the development of more complex circuits.

R#3_Q3: the authors should avoid using such superlative subjective terms like 'excellent'.

R#3_A3: We have now modified the text according to the Reviewer's comment.

REVIEWERS' COMMENTS:

Reviewer #1 (Remarks to the Author):

The authors have put a significant amount of effort into responding to comments. I have revisited the article to examine further the novelty and impact produced by the results from the "device physics and engineering point of view" as requested by the authors. Unfortunately, I still feel that the work has not made an advance of significance which is required for publication in Nature Communications. I would like to use this opportunity to express to the authors my rationale behind reaching this opinion based on the two degrees of novelty that the authors highlighted so that the review is fair and justified.

"i. the fabrication of printed Field Effect Transistors on paper, working with low supply voltages (smaller than 2 V), and with notable electrical performance (low threshold voltage and large mobility)."

If I examine the article from device physics and performance perspective, "Field Effect Transistors on paper, working with low supply voltages (smaller than 2 V), and with notable electrical performance (low threshold voltage and large mobility)" has already been demonstrated by S. Park and D. Akinwande, First demonstration of high performance 2D monolayer transistors on paper substrates, IEEE International Electron Devices Meeting (IEDM), 2017 (reference 18, now 25). Even with the authors ITRS definition of on/off ratio, the devices presented in this paper still only have comparable performance to state of the art.

The work presented by the authors has an inkjet-printed dielectric and electrode components however the performance (i.e. mobility and on/off ratio) of a FET is primarily attributed to the semiconducting channel which in this case, is not printed. Therefore it is essential to compare the device's performance to literature where the semiconducting channel is prepared by CVD material rather than devices with inkjet-printed channels.

In my opinion, the use of both paper and polymer substrates is primarily driven by the unique selling point of a flexible device. Therefore I disagree that a "direct comparison is misleading" (#A7). For figure S5 and S8 the authors highlight (#A7) that "despite being printed on paper, show performance comparable or even better than those of other devices fabricated on other flexible substrates", however, the devices only show performance comparable, better but also worse than other devices fabricated on flexible substrates which in my opinion is not a significant performance advance.

"ii. the choice of pre-patterned CVD MoS₂ as semiconducting layers deposited on paper, which gives the possibility to design and fabricate "on-demand" devices and circuits through a maskless fabrication technique such as inkjet printing. Here, in particular, we propose an ASIC design approach that we define "channel array", echoing the well-known "gate array" approach in the Electrical Engineering community, where only the channel is pre-defined, and all the other elements such as contacts, dielectric, and connections represent a degree of freedom for the designer. This has never been proposed before (especially on paper) and clearly is far from being incremental."

When I examine the engineering and fabrication of the devices through a combination of inkjet printing and CVD technology, the concept has already been achieved. Kim, et al. ACS Nano 2017, 11, 10, 10273-10280 (doi.org/10.1021/acsnano.7b04893) has used CVD grown MoS₂ with inkjet-printed PVP dielectric and PEDOT contacts to build a transistor on a flexible substrate (PEN).

Furthermore, the patterning of the CVD material into the design of the "channel array" has been attempted in a similar fashion. Kim, et al. ACS Nano 2017, 11, 10, 10273-10280 (doi.org/10.1021/acsnano.7b04893) has patterned CVD MoS₂ to create an array of transistor

channels before “all the other components are deposited using inkjet printing” (i.e. dielectric and contacts) which was all done on a flexible substrate. The engineering from a semi-printed CVD MoS₂ device in ambient conditions on a flexible substrate to a semi-printed CVD MoS₂ device in ambient conditions on paper substrate is not enough to merit a significant engineering advance in my opinion.

(R#1_A5): “Our work does not show conceptually new devices, as also stated by the Reviewer, but a new way to fabricate devices that does not lower their performance, where compared to devices fabricated through complex micro-fabrication techniques. ”

Please see my above points regarding novelty from both engineering and device performance perspectives.

(R#1_A5):

“As underlined in the main manuscript (page 3), Lin et al. (Lin, Z. et al. Solution-processable 2D semiconductors for high-performance large-area electronics. Nature 562, 254–258 (2018)), Ref. 19, now Ref. 27) reported transistors fabricated with solution-processed MoS₂, which show remarkable performance (average mobility of around 7–11 cm² V⁻¹ s⁻¹). However, the MoS₂ solution was deposited by spin coating and the electrical characterisation presented is relative to the devices fabricated on rigid SiO₂/Si substrate (as shown in Extended Data Fig. 8 of the manuscript), and not on Kapton. In addition, the fabrication steps require acid cleaning and annealing above 300 °C, which are incompatible with substrates such as paper, which is our target substrate.”

As stated in my original response Lin, Z. et al. also uses flexible Kapton substrates, as seen in figure 3i of their main text which makes the work relevant. They reach a comparable performance with a liquid exfoliated material when compared to the CVD material in this work. Therefore reviewer #3 point makes a lot of sense, why not just use a fully printed solution on a flexible substrate?

(R#1_A10):

We thank the Reviewer for his/her comment, as it gives us the opportunity to provide more details regarding the novelty of our work, which was clearly not well delivered, based on the Reviewer comment. The rationale behind our channel array system is the following: whilst the choice of a cheap, additive, mask-less technique such as inkjet printing for the development of a paper based electronic system might be obvious, the introduction of the more expensive CVD-grown semiconductor layers could be seen as counter-productive. Inkjet-printing, however, presents critical issues that have, so far, limited the development of high-quality semiconducting layers, which are necessary in any transistor structure. So far, the best field-effect mobility and ION/IOFF ratio reported for a TMD ink-based transistor are around 10 cm² V⁻¹ s⁻¹–110 cm²/Vs and (ION/IOFF)/VDD around 100, respectively, but through solutionprocess methods and not inkjet (Lin, Z. et al. Solution-processable 2D semiconductors for highperformance large-area electronics. Nature 562, 254–258 (2018)), which further need post-processing incompatible with paper substrate. Hence, until the issues associated with fully solution-process printed transistors are not solved, an alternative fabrication technique is strongly needed.

I don't understand how inkjet printing could present critical issues towards the development of high-quality semiconducting layers? I assume the authors mean that exfoliation of semiconducting layers in liquid has presented critical issues that have limited the development of high-quality semiconducting layers. I would have agreed until Lin, Z. et al solved a long-standing issue of solution-processed semiconducting material quality. Their process may be incompatible with a paper substrate, but it is compatible with other flexible substrates. Therefore it might not be timely anymore to consider an alternative semi-printed CVD fabrication technique as a significant advance towards high-performance flexible devices.

(R#1_Q10)

The combination of these methods, although well developed as individual techniques, is, however, far from trivial: the optimisation of the ink formulation and rheological properties, and the control of the interaction between the ink and the substrate, do not necessarily result in a successful print. Similarly, applying the printing parameters, which were found to be the optimised values for small films, does not automatically result in a good film formation for crystalline materials. Moreover, printing a liquid dielectric on CVD material may affect the transport properties of the crystalline channel.

The authors are correct, it is not trivial to optimise. However, Kim, et al. ACS Nano 2017 has already optimised complementary CVD-inkjet technology, reducing impact.

(R#1_Q10)

"The CVD method is currently expensive, though our approach is cheaper than conventional CVD requiring only a quartz tube oven (which costs in the region of €20,000) and solid precursors (Dumcenco, D. et al. Large-Area Epitaxial Monolayer MoS₂. ACS Nano 9, 4611–4620 (2015)), and may become the only solution if the issues associated to printed 2D transistors cannot be solved. Up to now, CVD is surely the most promising bottom up method for the large area synthesis of high-quality TMDs, especially considering the recent progress in the CVD growth of MoS₂ involving a low-cost, large-area roll-to-roll approach (Lim, Y. R. et al. Roll-to-Roll Production of Layer-Controlled Molybdenum Disulfide: A Platform for 2D Semiconductor-Based Industrial Applications. Adv. Mater. 13 30, 1705270 (2018)). It allows control of the layer number, thickness, domain size and morphology of the deposited layers. The fabrication approach demonstrated in our manuscript allows the manufacturing of printed FET transistors on paper, working with low supply voltages (smaller than 2 V), and with electrical performance comparable to those of flexible FETs developed using micro-fabrication techniques (see details on device performance provided above). Furthermore, it allows the "on demand" fabrication of more complex circuits through the ASIC design philosophy, providing full design flexibility of the circuit. Hence, we believe that our work does not deserve rejection only based on costs, which may change in future and may be justified by the final application. We would like to invite the Reviewer to also look at the novelty and impact produced by our results, both from the device physics and engineering point of views."

Please see my above point on Lin, Z. et al regarding high-quality solution-processed material.

I would like to emphasise here that I did not conclude rejection "only based on costs, which may change in future" but on the "on the basis of novelty and unremarkable performance" as stated in my first response. Please see above response to the two points the authors have identified as the novel aspects of the work from "the device physics and engineering point of views".

Reviewer #2 (Remarks to the Author):

The revision is acceptable. I recommend the publication of the manuscript.

Reviewer #3 (Remarks to the Author):

The authors have addressed the reviewer comments.

We thank the Reviewers for their comments. Point-by-point answers to their questions follow.

Reviewer #1 (Remarks to the Author):

R#1: The authors have put a significant amount of effort into responding to comments. I have revisited the article to examine further the novelty and impact produced by the results from the "device physics and engineering point of view" as requested by the authors. Unfortunately, I still feel that the work has not made an advance of significance which is required for publication in Nature Communications. I would like to use this opportunity to express to the authors my rationale behind reaching this opinion based on the two degrees of novelty that the authors highlighted so that the review is fair and justified.

We thank the Reviewer for his/her comments and for recognizing the work spent on the revision. We hope that in this second round we will be able to make him/her appreciate the value of our work.

R#1_Q1: "i. the fabrication of printed Field Effect Transistors on paper, working with low supply voltages (smaller than 2 V), and with notable electrical performance (low threshold voltage and large mobility)."

If I examine the article from device physics and performance perspective, "Field Effect Transistors on paper, working with low supply voltages (smaller than 2 V), and with notable electrical performance (low threshold voltage and large mobility)" has already been demonstrated by S. Park and D. Akinwande, First demonstration of high performance 2D monolayer transistors on paper substrates, IEEE International Electron Devices Meeting (IEDM), 2017 (reference 18, now 25). Even with the authors ITRS definition of on/off ratio, the devices presented in this paper still only have comparable performance to state of the art.

R#1_A1: We appreciate the value of the results reported in ref. 25 (Park, S. & Akinwande, D. First demonstration of high performance 2D monolayer transistors on paper substrates. in 2017 IEEE International Electron Devices Meeting (IEDM) 5.2.1-5.2.4 (IEEE, 2017)), however, we think that our work represents a step forward towards the development of paper-based electronics.

Here we want to remark again that our transistors exhibit a performance comparable to that reported in ref. 25, although fabricated with large-area, cost-efficient processes and using less mature, more challenging materials in ambient conditions. We also present a relatively large statistics of fabricated devices (over 26 devices). Moreover, the substrate we have adopted is more challenging with respect to the one used in ref. 25. As indeed underlined in Park's work, reducing the surface roughness is a crucial point for high yields and stable operation. For this reason, the glossy paper substrate used in that work is covered with a polyimide smoothing layer that confers a surface roughness of 3 nm, much smaller than that of our substrate (Figure S9). We would like to underline that using our commercial paper substrates without modifications is a further point of strength of our work. All this given, we do believe that our work is an improvement in paper electronics based on 2D materials.

R#1_Q2: The work presented by the authors has an inkjet-printed dielectric and electrode components however the performance (i.e. mobility and on/off ratio) of a FET is primarily attributed to the semiconducting channel which in this case, is not printed. Therefore it is essential to compare the device's performance to literature where the semiconducting channel is prepared by CVD material rather than devices with inkjet-printed channels.

R#1_A2: We agree with Reviewer #1 and indeed, as reported in our previous response and in the supplementary information (Figure S8 and Table S2), we have presented a very detailed comparison of the performance of our devices with that of other CVD-grown MoS₂ based FETs reported in the literature.

R#1_Q3: In my opinion, the use of both paper and polymer substrates is primarily driven by the unique selling point of a flexible device. Therefore I disagree that a "direct comparison is misleading" (#A7). For figure S5 and S8 the authors highlight (#A7) that "despite being printed on paper, show performance comparable or even better than those of other devices fabricated on other flexible substrates", however, the devices only show performance comparable, better but also worse than other devices fabricated on flexible substrates which in my opinion is not a significant performance advance.

R#1_Q3: As underlined in our previous reply, paper and plastic substrates are characterized by different surface chemistry (structure and energy), roughness, wettability, and flexibility, which eventually can lead to very different electrical properties of the materials: this is the reason why we stated that a direct comparison from a material point of view is misleading. Despite this, even when compared to devices fabricated on generic flexible substrates our results do generally exhibit better performance, which is far from being obvious, since paper is a truly challenging material, as explained above.

Moreover, using paper or paper-like substrates represents an advantage both from an economic and an environmental point of view. Paper is definitely one of the most used and cost-effective materials in daily life, with an average price close to 0.1 cent dm⁻² (lower than that of common plastic substrates such as polyethylene terephthalate (≈ 2 cent dm⁻²) and polyimide (≈ 30 cent dm⁻²). The amount of waste resulting from electrical and electronic equipment is expected to reach 52.3 million metric tons by 2021 (C.P. Baldé, *et al.* "The Global E-waste Monitor– 2017", International Solid Waste Association (ISWA), Bonn/Geneva/Vienna (2017)): introducing biocompatibility, reusability, and, eventually, biodegradability into the development of everyday electronics is thus unavoidable and will ease the recycling/disposal process and improve the cost-efficiency. Therefore paper substrates are attractive also from an economic point of view..

As stated in the paper, however, our devices have comparable performance to those reported for CVD-grown MoS₂ transistors fabricated on plastic substrates (Figure S8, purple dots).

R#1_Q4: "ii. the choice of pre-patterned CVD MoS₂ as semiconducting layers deposited on paper, which gives the possibility to design and fabricate "on-demand" devices and circuits through a mask-

less fabrication technique such as inkjet printing. Here, in particular, we propose an ASIC design approach that we define "channel array", echoing the well-known "gate array" approach in the Electrical Engineering community, where only the channel is pre-defined, and all the other elements such as contacts, dielectric, and connections represent a degree of freedom for the designer. This has never been proposed before (especially on paper) and clearly is far from being incremental."

When I examine the engineering and fabrication of the devices through a combination of inkjet printing and CVD technology, the concept has already been achieved. Kim, et al. ACS Nano 2017, 11, 10, 10273-10280 (doi.org/10.1021/acsnano.7b04893) has used CVD grown MoS₂ with inkjet-printed PVP dielectric and PEDOT contacts to build a transistor on a flexible substrate (PEN). Furthermore, the patterning of the CVD material into the design of the "channel array" has been attempted in a similar fashion. Kim, et al. ACS Nano 2017, 11, 10, 10273-10280 (doi.org/10.1021/acsnano.7b04893) has patterned CVD MoS₂ to create an array of transistor channels before "all the other components are deposited using inkjet printing" (i.e. dielectric and contacts) which was all done on a flexible substrate. The engineering from a semi-printed CVD MoS₂ device in ambient conditions on a flexible substrate to a semi-printed CVD MoS₂ device in ambient conditions on paper substrate is not enough to merit a significant engineering advance in my opinion.

(R#1_A5): "Our work does not show conceptually new devices, as also stated by the Reviewer, but a new way to fabricate devices that does not lower their performance, where compared to devices fabricated through complex micro-fabrication techniques. "

Please see my above points regarding novelty from both engineering and device performance perspectives.

R#1_A4: We thank the Reviewer for the valuable comment and indeed we do appreciate the value of the work presented in Kim, T. Y. et al. (Transparent Large-Area MoS₂ Phototransistors with Inkjet-Printed Components on Flexible Platforms. ACS Nano 11, 10273–10280 (2017)), which has been referenced in the main text (ref. 30), when describing the CVD + inkjet technology. The paper the Reviewer is mentioning is anyway not undermining the novelty and the value of the present work.

We want to stress that the technology we have developed is fully focused and oriented towards the realization of devices and integrated circuits printed on paper. This has been possible through the optimization of the process, in which CVD MoS₂ stripes at the millimeter scale have been successfully transferred on paper substrates, and novel hBN-based inks have been exploited, eventually allowing the fabrication of devices with performance comparable to what is obtained in the literature using micro-fabrication techniques over flexible substrates. As pointed out by the authors in ref. 30, the use of organic materials (characterized by low conductivity and low relative permittivity) negatively affected the electrical performance of the CVD-grown MoS₂ phototransistors fabricated on plastic substrates, which exhibited a mobility equal to 0.37 cm² V⁻¹ s⁻¹ and an I_{ON}/I_{OFF} ratio around 10², while requiring

relatively large supply voltages. Looking at our results, we did indeed optimize the use of CVD and inkjet-printing, in terms of materials, fabrication processes and achieved device performance.

R#1_Q5: (R#1_A5) : "As underlined in the main manuscript (page 3), Lin et al. (Lin, Z. et al. Solution-processable 2D semiconductors for high-performance large-area electronics. Nature 562, 254–258 (2018)), Ref. 19, now Ref. 27) reported transistors fabricated with solution-processed MoS₂, which show remarkable performance (average mobility of around 7–11 cm² V⁻¹ s⁻¹). However, the MoS₂ solution was deposited by spin coating and the electrical characterization presented is relative to the devices fabricated on rigid SiO₂/Si substrate (as shown in Extended Data Fig. 8 of the manuscript), and not on Kapton. In addition, the fabrication steps require acid cleaning and annealing above 300 °C, which are incompatible with substrates such as paper, which is our target substrate."

As stated in my original response Lin, Z. et al. also uses flexible Kapton substrates, as seen in figure 3i of their main text which makes the work relevant. They reach a comparable performance with a liquid exfoliated material when compared to the CVD material in this work. Therefore reviewer #3 point makes a lot of sense, why not just use a fully printed solution on a flexible substrate?

R#1_A5: We thank the Reviewer for the comment. Actually, our target substrate is paper and the process described in Lin *et al.* (Lin, Z. et al. Solution-processable 2D semiconductors for high-performance large-area electronics. Nature 562, 254–258 (2018)), Ref. 27), despite being a breakthrough in the field and really relevant, is not compatible with the thermal budget of our substrate. Regarding our previous comments, we want to apologize if we were not sufficiently clear. In particular, we did not mean that no results on Kapton are reported in the paper, but that the detailed electrical characterization (mobility and I_{ON}/I_{OFF} values) is referred to devices fabricated on rigid SiO₂/Si substrates; we definitively thank the Reviewer for pointing this out. Once again, we would like to stress that, in Lin *et al.*, the result on Kapton can be considered only as a side result, since just one characteristic is shown, and no in-depth analysis is performed.

R#1_Q6: (R#1_A10:) We thank the Reviewer for his/her comment, as it gives us the opportunity to provide more details regarding the novelty of our work, which was clearly not well delivered, based on the Reviewer comment. The rationale behind our channel array system is the following: whilst the choice of a cheap, additive, mask-less technique such as inkjet printing for the development of a paper based electronic system might be obvious, the introduction of the more expensive CVD-grown semiconductor layers could be seen as counter-productive. Inkjet-printing, however, presents critical issues that have, so far, limited the development of high-quality semiconducting layers, which are necessary in any transistor structure. So far, the best field-effect mobility and I_{ON}/I_{OFF} ratio reported for a TMD ink-based transistor are around 10 cm² V⁻¹ s⁻¹ and $(I_{ON}/I_{OFF})/V_{DD}$ around 100, respectively, but through solution process methods and not inkjet (Lin, Z. et al. Solution-processable 2D semiconductors for high performance large-area electronics. Nature 562, 254–258 (2018)), which further need post-processing incompatible with paper substrate. Hence, until the issues associated with

fully solution-process printed transistors are not solved, an alternative fabrication technique is strongly needed.

I don't understand how inkjet printing could present critical issues towards the development of high-quality semiconducting layers? I assume the authors mean that exfoliation of semiconducting layers in liquid has presented critical issues that have limited the development of high-quality semiconducting layers. I would have agreed until Lin, Z. et al solved a long-standing issue of solution-processed semiconducting material quality. Their process may be incompatible with a paper substrate, but it is compatible with other flexible substrates. Therefore it might not be timely anymore to consider an alternative semi-printed CVD fabrication technique as a significant advance towards high-performance flexible devices.

R#1_A6: As underlined by the reviewer, solution processes of 2D materials (including solution-phase exfoliation, ion/molecule intercalation and exfoliation, and wet-chemical synthesis) are still less mature techniques if compared to the most commonly employed mechanical exfoliation or chemical vapour deposition. The work reported by Lin *et al.* (Solution-processable 2D semiconductors for high-performance large-area electronics. Nature 562, 254–258 (2018), ref. 27) does indeed represent a breakthrough in the field. Another interesting example was reported in 2018 by Yang *et al.* (Yang, S. et al. A Delamination Strategy for Thinly Layered Defect-Free High-Mobility Black Phosphorus Flakes. Angew. Chemie Int. Ed. 57, 4677–4681 (2018)) who demonstrated solution-processed phosphorene FETs obtained by delamination of bulk black phosphorous using an electrochemical strategy. Single-flake devices showed an average I_{ON}/I_{OFF} ratio of $\sim 10^4$ and a hole mobility of $\sim 200 \text{ cm}^2 \text{ V}^{-1} \text{ s}^{-1}$ at room temperature. However, both in Lin's and Yang's papers, the semiconductors were deposited using a spin coating technique. As it is widely reported in the literature, (Sahu, N., Parija, B. & Panigrahi, S. Fundamental understanding and modeling of spin coating process: A review. Indian J. Phys. 83, 493–502 (2009)), spin coating cannot be considered a large area process as its main disadvantages are poor materials efficiency (and subsequent reduced cost efficiency), lack of patterning capability, and the fact that large substrates cannot be spun at a sufficiently high rate in order to allow the deposition of thin films.

One of the major obstacles to the development of all inkjet-printed FET is related to the fact that, once a suitable solution-processable 2D semiconductor is selected, developing a stable and printable ink capable to retain the intrinsic material properties becomes the key factor. In order to formulate an ink, surfactants, solvents, or carrier liquids are always required. Solubility, surface energy, boiling point, stability, viscosity, and orthogonality of the solvents are some of the chemical and physical properties together with roughness, uniformity and continuity that should be taken into account. Exhaustive discussions about the optimization of the ink formulations and the printing processes and the relative problematics can be found in Li, J., Naiini, M. M., Vaziri, S., Lemme, M. C. & Östling, M. Inkjet

Printing of MoS₂. *Adv. Funct. Mater.* **24**, 6524–6531 (2014), McManus, D. et al. Water-based and biocompatible 2D crystal inks for all-inkjet-printed heterostructures. *Nat. Nanotechnol.* **12**, 343–350 (2017), and Seo, J.-W. T. et al. Fully Inkjet-Printed, Mechanically Flexible MoS₂ Nanosheet Photodetectors. *ACS Appl. Mater. Interfaces* **11**, 5675–5681 (2019). As shown by Li et al., for example, although MoS₂ is intrinsically an n-type semiconductor, solution processing can introduce unintentional dopants or traps which may influence the electronic properties of the material, giving a p-type behaviour. Furthermore, due to the typically small lateral size of solution processed 2D semiconductors, inkjet-printed FETs are generally based on network of flakes channels, which show smaller mobilities if compared to single flake counterparts, due to scattering at flake edges, and to intra-flake hopping mechanisms. Kelly et al. reported mobility values in the range from 0.08 to 0.22 cm² V⁻¹ s⁻¹ for multi-flake inkjet-printed channel FETs (prepared by means of liquid-phase exfoliation of various TMDCs in N-methyl 2-pyrrolidone), whilst intrinsic nanosheet mobilities determined by optical-pump terahertz probe spectroscopy were between 47 and 91 cm² V⁻¹ s⁻¹ (Kelly, A. G. et al. All-printed thin-film transistors from networks of liquid-exfoliated nanosheets. *Science (80-.).* **356**, 69–73 (2017), ref. 20).

Thus, considering the state-of-the-art and the device performances obtained in the present work, we do believe we have demonstrated that, for the time being, a semi-printed CVD fabrication approach is the most effective strategy to obtain high-performance and low-cost flexible devices.

R#1_Q7: (R#1_Q10) *The combination of these methods, although well developed as individual techniques, is, however, far from trivial: the optimisation of the ink formulation and rheological properties, and the control of the interaction between the ink and the substrate, do not necessarily result in a successful print. Similarly, applying the printing parameters, which were found to be the optimised values for small films, does not automatically result in a good film formation for crystalline materials. Moreover, printing a liquid dielectric on CVD material may affect the transport properties of the crystalline channel.*

The authors are correct, it is not trivial to optimise. However, Kim, et al. ACS Nano 2017 has already optimised complementary CVD-inkjet technology, reducing impact.

R#1_A7: Please refer to R#1_A4.

R#1_Q8: (R#1_Q10) *"The CVD method is currently expensive, though our approach is cheaper than conventional CVD requiring only a quartz tube oven (which costs in the region of €20,000) and solid precursors (Dumcenco, D. et al. Large-Area Epitaxial Monolayer MoS₂. ACS Nano 9, 4611–4620 (2015)), and may become the only solution if the issues associated to printed 2D transistors cannot be solved. Up to now, CVD is surely the most promising bottom up method for the large area synthesis of high-quality TMDs, especially considering the recent progress in the CVD growth of MoS₂ involving a*

low-cost, large-area roll-to-roll approach (Lim, Y. R. et al. Roll-to-Roll Production of Layer-Controlled Molybdenum Disulfide: A Platform for 2D Semiconductor-Based Industrial Applications. Adv. Mater. 13 30, 1705270 (2018)). It allows control of the layer number, thickness, domain size and morphology of the deposited layers. The fabrication approach demonstrated in our manuscript allows the manufacturing of printed FET transistors on paper, working with low supply voltages (smaller than 2 V), and with electrical performance comparable to those of flexible FETs developed using micro-fabrication techniques (see details on device performance provided above). Furthermore, it allows the "on demand" fabrication of more complex circuits through the ASIC design philosophy, providing full design flexibility of the circuit. Hence, we believe that our work does not deserve rejection only based on costs, which may change in future and may be justified by the final application. We would like to invite the Reviewer to also look at the novelty and impact produced by our results, both from the device physics and engineering point of views."

Please see my above point on Lin, Z. et al regarding high-quality solution-processed material.

I would like to emphasize here that I did not conclude rejection "only based on costs, which may change in future" but on the "on the basis of novelty and unremarkable performance" as stated in my first response. Please see above response to the two points the authors have identified as the novel aspects of the work from "the device physics and engineering point of views".

R1#_A8: We thank the Reviewer for agreeing on the issue of the cost of the present technology. We want anyway to remark that we believe that the novelty and the impact of the present work has already been discussed in our previous reply (and again remarked in R#1_A4 and R#1_A6), as also recognized by Reviewers #2 and #3.

The approach proposed in our work allows the development of "on demand" circuits using an ASIC design philosophy (thus providing full design flexibility of the circuit), exploiting CVD-grown semiconducting substrates in combination with an inkjet-printing technique to fabricate functional thin film devices on paper substrates. Printed FETs on paper, operating at low supply voltages (smaller than 2 V) and with notable electrical performance (low threshold voltage and high mobility), comparable to those of flexible FETs developed using micro-fabrication techniques and high performing inorganic oxides, are presented. As underlined in the previous reply, the reported performance does not only depend on the use of a high-quality semiconductor film, but on the adoption of high-quality hBN inkjet-printed layers as well. This is a point of strength of our work.

We also believe that the obtained results are far from being "unremarkable", as we believe we have shown through comparison with devices available in the literature and on substrates less challenging than paper. We would like to stress again that we do believe that our results represent an improvement in paper-based electronics, both from the points of view of fabrication and of achieved performance, and should be considered as starting points towards the development of more complex circuits.

Reviewer #2 (Remarks to the Author):

R#2: The revision is acceptable. I recommend the publication of the manuscript.

We really thank the Reviewer for the appreciation of our work and for recommending the publication of the manuscript.

Reviewer #3 (Remarks to the Author):

R#3: The authors have addressed the reviewer comments.

We really thank the Reviewer for her/his positive comment on the revised version of the manuscript.